# Discrete photoentrainment of mammalian central clock is regulated by bi-stable dynamic network in the suprachiasmatic nucleus

Po-Ting Yeh[1,2], Kai-Chun Jhan[3], Ern-Pei Chua[1], Wun-Ci Chen[3], Shi-Wei Chu [4,5], Shun-Chi Wu[3,5] & Shih-Kuo Chen [1,6,7] ✉

The biological clock synchronizes with the environmental light-dark cycle through circadian photoentrainment. While intracellular pathways regulating clock gene expression after light exposure in the suprachiasmatic nucleus are well studied in mammals, the neuronal circuits driving phase shifts remain unclear. Here, using a mouse model, we show that chemogenetic activation of early-night light-responsive neurons induces phase delays at any circadian time, potentially breaking the photoentrainment dead zone. In contrast, activating late-night light-responsive neurons mimics light-induced phase shifts. Using in vivo two-photon microscopy, we found that most neurons in the suprachiasmatic nucleus exhibit stochastic light responses, while a small subset is consistently activated in the early subjective night and another is inhibited in the late subjective night. Our findings suggest a dynamic bi-stable network model for circadian photoentrainment, where phase shifts arise from a functional circuit integrating signals to groups of outcome neurons, rather than a labeled-line principle seen in sensory systems.

The circadian clock controls many physiological functions in animals, including gene expression, stem cell proliferation, metabolism, neuronal activity, gut microbiota, mood, learning, and memory[1]. In mammals, the suprachiasmatic nucleus (SCN) in the hypothalamus serves as the master clock, orchestrating many peripheral clocks within different tissues[2–5]. It is essential for these organisms to have an endogenous circadian clock that can be entrained by external environmental cues. The SCN receives direct inputs from a unique group of retinal neurons, the intrinsically photosensitive retinal ganglion cells (ipRGCs), via the retinohypothalamic tract (RHT), conveying environmental luminance information[6–17]. Consequently,

the light pulse signal from ipRGCs is crucial for mammals, such as mice and humans, to synchronize their internal clocks with the external light-day cycle, a process known as circadian photoentrainment[17–21]. Different species all demonstrate discrete light responses for circadian photoentrainment. Specifically, light exposure during the early subjective night causes a phase delay, while exposure during the late subjective night results in a phase advance[22–24]. Moreover, during the 'dead zone' in the middle of the subjective day, light produces only minimal phase shifts[25]. However, fundamental properties in the central clock of animals that produce phase-shift behavior outputs remain controversial.

[1]Department of Life Science, National Taiwan University, Taipei 10617, Taiwan. [2]Taiwan International Graduate Program in Interdisciplinary Neuroscience, National Taiwan University and Academia Sinica, Taipei 11529, Taiwan. [3]Department of Engineering and System Science, National Tsing Hua University, Hsinchu 30013, Taiwan. [4]Department of Physics, National Taiwan University, Taipei 10617, Taiwan. [5]Brain Research Center, National Tsing Hua University, Hsinchu 30013, Taiwan. [6]Neurobiology and Cognitive Science Center, National Taiwan University, Taipei 10617, Taiwan. [7]Center for Biotechnology, National Taiwan University, Taipei 10617, Taiwan. ✉e-mail: alenskchen@ntu.edu.tw

Recent studies across various model organisms indicate that circadian photoentrainment is regulated by diverse mechanisms. In insects, light directly reaches clock neurons in the brain, allowing cryptochrome or rhodopsin within these neurons to modulate the period-timeless complex and thereby reset the transcription-translation feedback loop (TTFL) in response to external light cues[26–28]. In contrast, in mammals, ipRGC input to the SCN is required for photoentrainment[18,20,21]. Light can modulate TTFL components, such as Per1, through glutamate and pituitary adenylate cyclase-activating polypeptide (PACAP) released by ipRGCs[29–34]. While light can induce Per1 expression to daytime levels at many circadian times, this alone does not fully explain mammalian circadian photoentrainment. For example, during the daytime when light-induced phase shift is limited, high-intensity light can still modulate activities or calcium responses within the SCN neurons[35–37]. Additionally, under distinct light-dark cycles, the TTFL coupling and epigenetic profiling of SCN neurons can be dynamic[38,39]. Finally, light exposure at different times of day could produce distinct clock genes or immediate early gene expression patterns[30,40]. Although the molecular mechanism of phase shift is extensively studied and some mathematical models have been proposed[41–44], the specific intra-SCN neuronal circuits that are involved in circadian photoentrainment remain poorly understood.

The SCN in mammals is a compact nucleus, composed of many cell types expressing different neurotransmitters with complex intra-nucleus network[45]. Recent single-cell transcriptomic analysis from the mouse SCN suggests it can be divided into at least five distinct molecular subtypes[46,47], including vasoactive intestinal peptide (VIP) and gastrin-releasing peptide (GRP) expressing neurons in the "core" and arginine vasopressin (AVP) expressing neurons in the "shell" regions[48]. Classic models have suggested a linear information processing pathway within the SCN. For example, the SCN core region, including VIP and GRP neurons, shows a higher amount of cFos expression than the shell region after light pulses[40,49–51]. Exogenous application of VIP or GRP can produce circadian phase shifts both in vivo and in vitro[52–55]. Furthermore, silencing the activity of VIP neurons could diminish light-induced phase shift[36]. Together, these results suggest luminance information is received by VIP and GRP neurons. However, other evidence suggests that the SCN comprises complex networks. For instance, optogenetic activation of VIP neurons can induce phase delay at early subjective night but does not produce the full PRC equivalent to the light pulse[37]. Conversely, activation of cholecystokinin (CCK) neurons within the SCN can only elicit phase advances during the late subjective night[56]. Thus, a time-gated functional network could be the underlying mechanism for the discrete properties of the PRC. Given the large number of neurons in the mammalian central clock, it is likely that more complex neuronal interactions and computations are involved in the circadian phase response processes. Consequently, population-wide recording of SCN neurons with single-cell resolution in vivo is crucial to deciphering the functional network and the neuronal mechanisms behind the circadian photoentrainment.

To investigate the time-gated properties of the SCN we modulate neuronal activity using designer receptors exclusively activated by designer drugs (DREADDs)[57] specifically in circadian time (CT) 16 or CT 22 light-responsive SCN neurons with the targeted recombination in the active population (TRAP) mouse system[58,59]. Activation of neurons trapped at CT 16 could induce phase delays throughout the day, disrupting the photoentrainment dead zone, while activation of neurons trapped at CT 22 could induce phase shifts similar to light. These results suggest there are multiple functional distinct populations in the SCN. Next, to further elucidate the distinct functional population in the single-cell resolution, we utilized in vivo two-photon microscopy with gradient-index (GRIN) endoscopes to record individual SCN neuronal activity using the genetically encoded calcium indicator GCaMP7f[60,61]. Our findings reveal that SCN neurons display time-gated dynamic network properties, with a small subset of neurons showing consistent light responses at zeitgeber time (ZT) 16 and ZT 22, aligning with circadian phase delay and advance, respectively. Our work suggests that information processing in the SCN likely comprises a bi-stable population-wide dynamic network rather than a linear hierarchical circuitry for circadian photoentrainment.

## Results

### Activation of CT 16-trapped light-responsive neurons produces phase delay and breaks circadian photoentrainment dead zone

To test whether SCN neurons utilize specific neuronal circuitry for discrete circadian photoentrainment, we used TRAP2 mice (Fos[2A-iCreER/2A-iCreER]) with Cre-dependent AAV containing excitatory Gs-coupled DREADDs (rM3Ds) to target neurons within the SCN that are responsive to light at CT 16 or CT 22 (Fig. 1a). Four weeks post-injection with either AAV9-hSyn-DIO-rM3D(Gs)-mCherry or AAV9-hSyn-DIO-EGFP into the SCN, we randomly divided the mice into two groups. Each group was subjected to a 900 lux, 10-minute light pulse at CT 16 or CT 22, followed by an injection of 4-OHT to label light-responsive SCN neurons as TRAP-CT 16 or TRAP-CT 22 (Fig. 1b and Supplementary Fig. 1). First, we confirmed that these TRAPed and control mice still showed normal light induced phase shift at CT 16, CT 22, and CT 8 respectively (Supplementary Fig. 2). We then administered CNO or PBS as a control at various times to assess whether the activation of TRAP-CT 16 or TRAP-CT 22 neurons could mimic the phase shifts typically induced by light (Fig. 1c–e, Supplementary Fig. 3 and 4). Given that CNO is known to affect DREADD neurons for 4-6 h post-injection, we strategically administered it at CT 2 to target the dead zone. Remarkably, the activation of TRAP-CT 16 neurons led to a significant phase delay across all three time points, including the photoentrainment dead zone and advance zone, in contrast to control mice injected with CNO (Fig. 1f). Analysis of phase shift using acrophase showed that CNO injection at CT 22 or CT 2 can produce phase delay in 4 out of 6 or 5 out 6 TRAP-CT16 mice respectively (Supplementary Fig. 3). Notably, activation of TRAP-CT 22 neurons resulted in a significant phase delay at CT 16, a significant phase advance at CT 22, and no phase shift at CT 2, similar to the usual light-induced phase response curve (Fig. 1g). To eliminate the potential effect of endogenous cFos expression in SCN neurons at CT 16, we also performed TRAPing without a light pulse. We confirmed that dark-TRAP at CT 16 could not produce a phase shift when CNO was injected at CT 16 (Supplementary Fig. 5). This result suggests the presence of at least two functional light-responsive circuits in the SCN: a dynamic CT 22-trapped circuit that can produce multiple behavioral responses depending on the circadian time, and a phase delay CT 16-trapped circuit that can be isolated to generate a phase delay regardless of circadian time, even during the photoentrainment dead zone.

### In vivo Two-Photon Calcium Imaging Reveals Diverse and Dynamic Light Responses in SCN Neurons

The test whether these are specific functional circuits within the SCN at distinct circadian times for photoentrainment, we employed two-photon microscopy with GRIN endoscopes in awake mice to observe real-time neuronal activity using calcium imaging in the SCN at the single-cell level across multiple times (Fig. 2a). VIP[Cre/+]; Ai14[CAG-tdTomato/+] mice were injected with AAV9-hSyn-GCaMP7f into the SCN (Fig. 2b and Supplementary Fig. 6a–b) and subsequently maintained under a 12:12 light-dark cycle for recordings at ZT 8, ZT 16, and ZT 22. This in vivo setup allows us to track single neurons across different ZT time points longitudinally (Fig. 2c). Furthermore, the two-photon system also allows us to image across several hundred microns in the dorsoventral axis (Fig. 2e) compared to single-photon systems[62]. Here, each recording session consisted of three trials of a 30-second dark baseline followed by a 90-second light exposure (488 nm, 1.76 μW/mm2) to the eyes. Activity from two z-positions within the SCN, separated by 105 μm, was captured during the same session (Fig. 2e). To enhance the

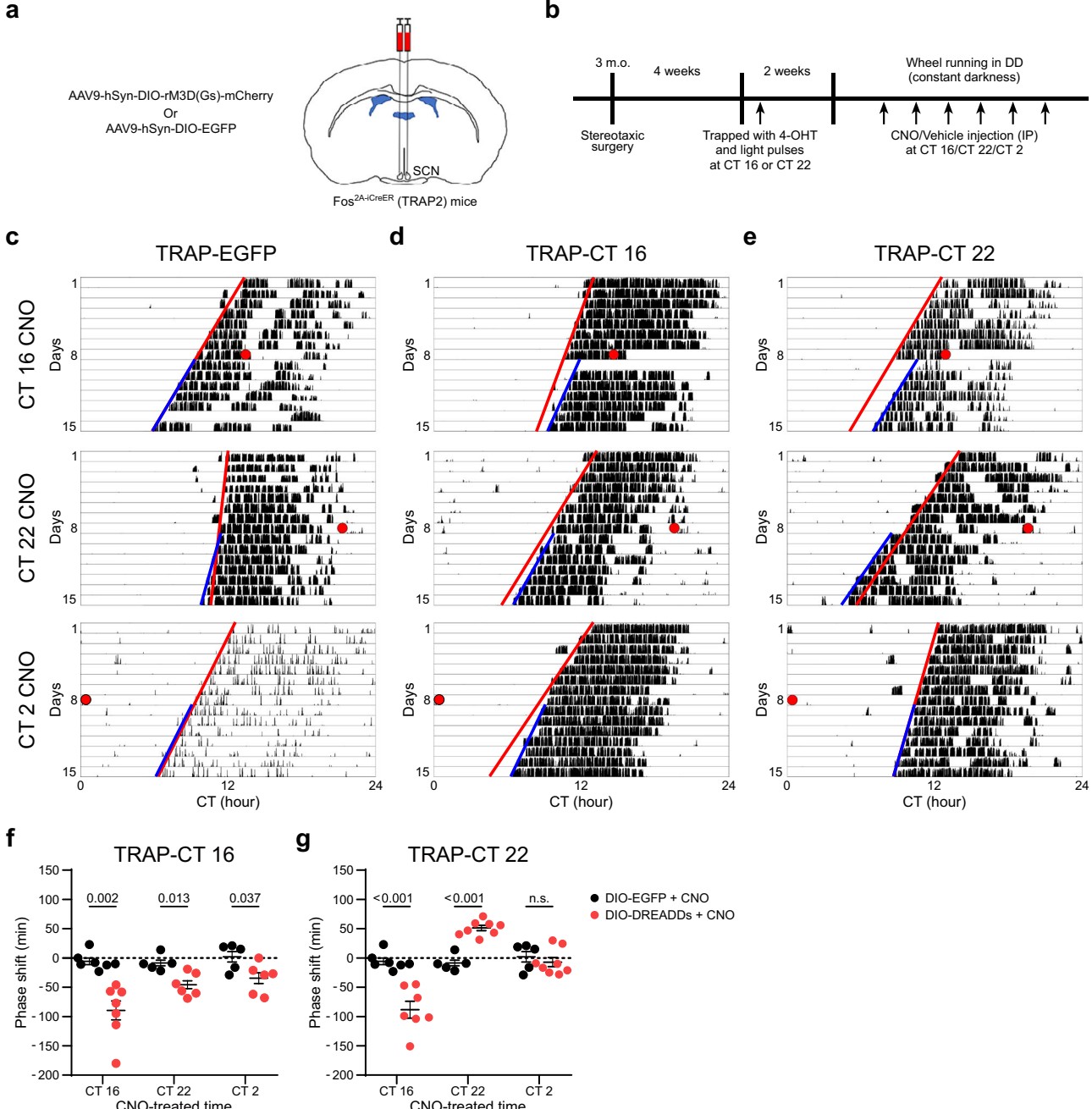

**Fig. 1 | Activation of CT 16-trapped light-responsive neurons produces phase delay and breaks the circadian photoentrainment dead zone. a** Experimental scheme for DREADDs bilateral SCN injection in the TRAP2 (Fos-iCreER) mice. **b** Time table for the experimental procedure. **c.** Representative actogram for GFP-expressing CT 16 or CT 22-trapped mice before and after CNO injection. **d** Representative actogram for DREADDs (rM3Ds)-expressing CT 16-trapped mice before and after CNO injection. **e** Representative behavioral data for DREADDs (rM3Ds)-expressing CT 22-trapped mice before and after CNO injection. The red dots represent the time points of CNO injection, while the red line depicts an extended linear regression based on activity onsets prior to the treatment. The blue lines reflect the activity onsets following treatment. **f** Statistics of phase shift analysis for DREADDs (rM3Ds)-expressing CT 16-trapped mice. Here, the phase shift form TRAP-CT16 mice injected with CNO were calculated using onset with best of fit line generated according to 3–5 days prior to injection. **g** Statistics of phase shift analysis for DREADDs (rM3Ds)-expressing CT 22-trapped mice. Numbers indicate adjust $p$ value and n.s. indicates $p > 0.05$, calculated by Holm-Šídák multiple t-test. $n = 7/7, 5/6, 5/6$ as control/DREADDs at CT 16, CT 22, CT 2 respectively for **f**. $n = 7/7, 5/8, 5/8$ as control/DREADDs at CT 16, CT 22, CT 2 respectively for g. Error bars indicate mean with SEM. Outline of mouse brain in a was drawn according to Paxinos and Franklin's the Mouse Brain[79].

statistical power, we repeated the same recording procedure for each ZT time from 3 different days to increase the total number of activity traces. After motion correction and cell segmentation, we successfully identified 338 regions of interests (ROIs) from three mice across all sessions (Supplementary Fig. 6c–d). Among these ROIs, 113 neurons, including 46 tdTomato-positive VIP neurons (40.7%) were consistently registered across 27 sessions (Fig. 2f). Furthermore, of the neurons located in the ventral focus plane, 44.6% are VIP+, while in the dorsal focus plane, 33.3% are VIP+. Subsequent analysis of the average neuronal response indicated a modest, yet consistent, positive light-induced response among SCN neurons at all three ZTs (Fig. 2g), confirming prior photometry findings[37].

To dissect individual SCN neuronal light responses, we analyzed the Z-score of GCaMP signals from 113 neurons (Fig. 3a and

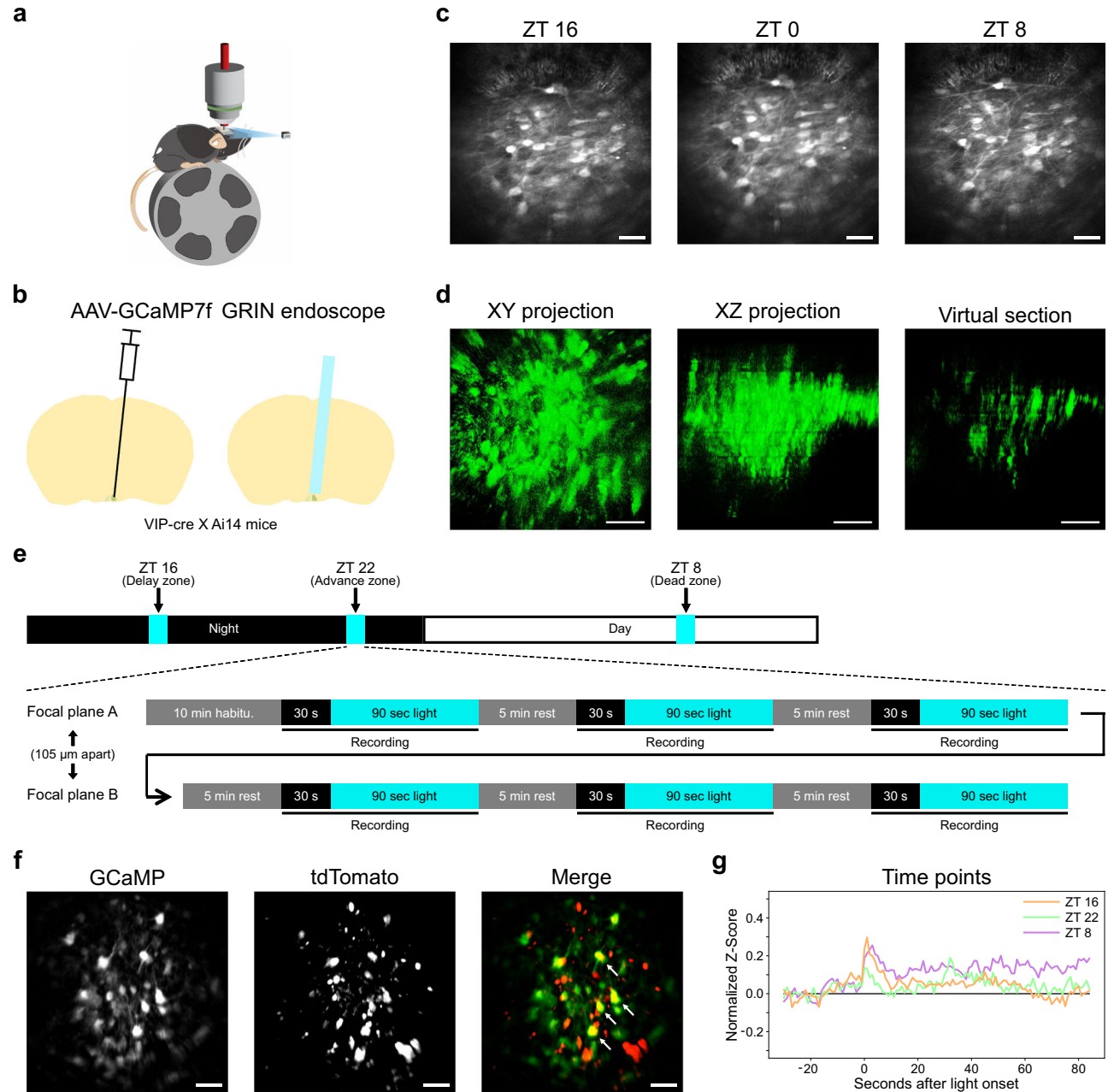

**Fig. 2 | Observation of SCN neuronal light responses in awake mice. a, b** An illustration of recording setup and surgical procedure. **c.** Representative two-photon in vivo GCaMP images (averaged) of SCN taken at different ZT times through the GRIN endoscope. **d** Representative 3D-projection of SCN GCaMP signal using z-stack images through the GRIN endoscope. The XY projection shows the summation of horizontal planes, while the XZ projection shows the summation of sagittal planes of the SCN. The XZ virtual 45-μm-thick section demonstrates the elongated point spread function in the z-axis through GRIN endoscopes. **e** Schemes for time points experiments (upper panel) and calcium imaging recording (lower panel). **f** Representative images of GCaMP and tdTomato signals collected simultaneously with fast-scanning two-photon system. Arrows indicate the colocalization of both channels. **g** Averaged light-response of identified neurons at 3 ZT times. Red: ZT 22, green: ZT 8, blue: ZT 16. *n* = 3 animals. Scale bars are 100 μm, according to the imaging side of the GRIN endoscope. Outline of mouse brain in b was drawn according to Paxinos and Franklin's the Mouse Brain[79].

Supplementary Fig. 7). Given the diversity of spontaneous calcium transients exhibited by SCN neurons, we averaged the Z-scores in 5-second bins to analyze light response trends for 90 sec. Subsequently, principal component analysis (PCA) was conducted on data from all trials of the 113 neurons, and k-means clustering was utilized to categorize the light responses into seven distinct clusters of light response (Fig. 3b). We plotted the mean light response for each cluster, including transient activation (cluster 1, 7.48%), sustained activation (cluster 2, 5.01%), delayed activation (cluster 3, 14.7%), marginal response (cluster 4, 35.5%), delayed inhibition (cluster 5,

16.1%), sustained inhibition (cluster 6, 4.42%), and transient inhibition (cluster 7, 16.8%) (Fig. 3c). Notably, neurons were not limited to a single type of light response; on average, each neuron exhibited 5.57 ± 1.08 (mean with SD) distinct types of light response across the 27 trials (Fig. 3d, e). Furthermore, even when using the most stringent criteria that consider marginal responses (cluster 4) as non-responsive to light, every neuron exhibited either activation (cluster 1-3) or inhibition (cluster 5-7) in at least 4 out of the total 27 trials (16%). These findings not only demonstrate the capability of SCN neurons to generate multiple types of light response but also imply

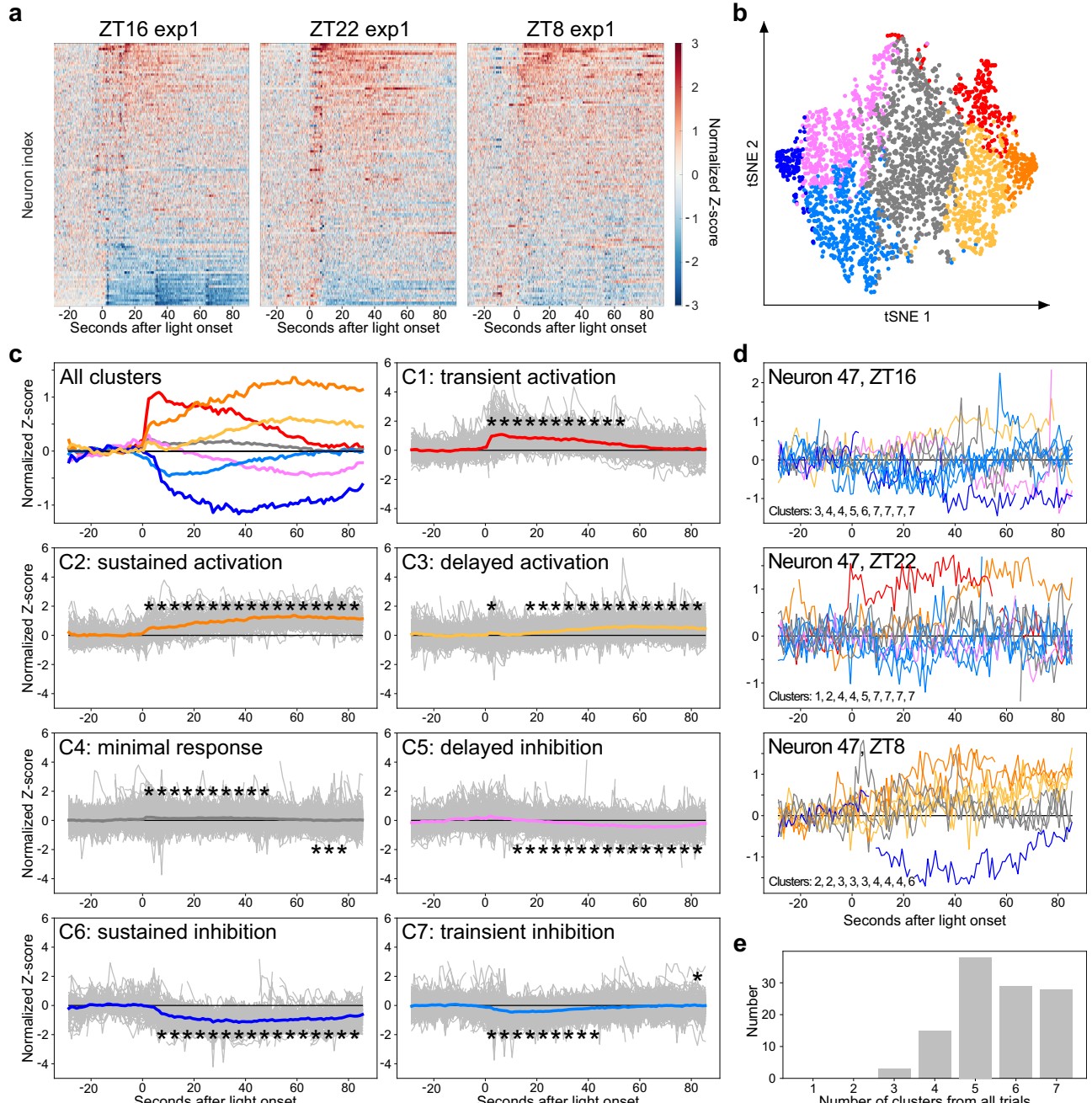

**Fig. 3 | Diverse neuronal light responses in the SCN. a** Representative heat map with GCaMP traces (normalized Z-score) from 113 identified neurons at ZT 16 (left), ZT 22 (middle), and ZT 8 (right), respectively. **b**, **c** 2D t-SNE map (**b**) shows the diversity of neuronal light-responses. 2875 recording traces from 113 neurons are plotted, excluding those with excessive motion artifacts. The 2875 recordings are classified into 7 clusters, indicated by different colors, with k-means. The 7 light responses are plotted in **c**, including averaged traces of all clusters (upper-left panel) and raw traces (gray) with average from each cluster (the other 7 panels). Stars above or below the average traces indicate significantly higher or lower z-score compared to the -5 - 0 second baseline, respectively. One-way ANOVA and Tukey post hoc test, * indicates $p < 0.05$. **d** Raw traces from the representative neuron that show high diversity of light-evoked responses through 9 repeated trials from 3 time points. **e** Histogram of neurons shows multiple types of light responses. $n = 3$ animals.

the presence of a dynamic functional circuitry involving most if not all SCN neurons to encode light signals.

## Functional Homogeneity in Light Response Patterns of VIP and Non-VIP SCN Neurons

The suprachiasmatic nucleus (SCN) is known to contain a diversity of neurons, including those expressing vasoactive intestinal peptide (VIP) and arginine vasopressin (AVP). Within this traditional molecular classification, VIP[+] and gastrin-releasing peptide (GRP[+]) neurons have

been viewed as the primary light reception unit in the SCN. To investigate the specificity of VIP neurons in light response relative to non-VIP neurons, we next analyzed the difference between tdTomato-positive (VIP[+]) and tdTomato-negative (VIP[-]) neurons. Contrary to expectations, our analysis revealed no significant differences in the average light responses between VIP[+] and VIP[-] neurons (Fig. 4a), nor the composition of their light responses (Fig. 4b). We further quantified the similarity of light response by computing Pearson correlation coefficients within and between VIP[+] and VIP[-] neurons in the same trial

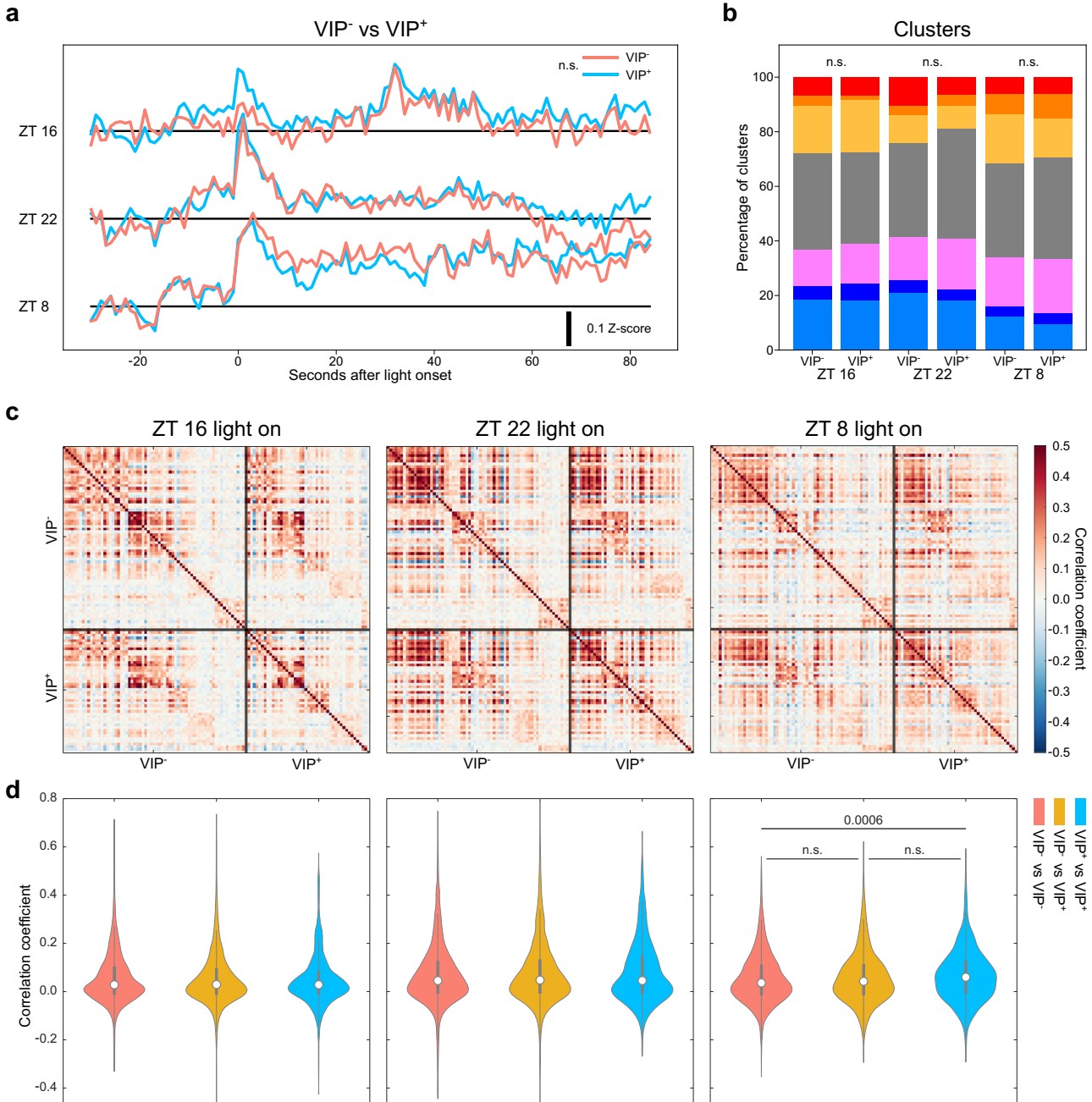

**Fig. 4 | VIPergic neurons show similar light responses to non-VIP neurons.**
**a** Averaged light response traces from VIP+ neurons (blue) and VIP- neurons (pink).
n.s. indicates $p > 0.05$ by two-way ANOVA Tukey post hoc tests. **b** Composition of 7
clusters of light response at different time points among VIP+ and VIP- neurons.
Color code follows Fig. 3c. No significantly difference is observed between VIP+ and
VIP- neurons (two-way ANOVA). **c** Correlation maps comparing VIP+ and VIP- neu-
rons. Each pixel on the map represents the Pearson correlation coefficient between
two neurons. **d** Violin plots summarize the Pearson correlation coefficient of each
comparison pair among cell types, including VIP- to VIP- (pink), VIP+ to VIP- (yellow),
and VIP+ to VIP+ (blue). The number indicates $p$ value, n.s. indicates $p > 0.05$, by one-
way ANOVA Tukey post hoc tests. $n = 3$ animals. For violin plots, circles indicate
median, thick vertical lines indicate interquartile ranges and thin vertical lines
indicate 1.5X interquartile range.

(Fig. 4c and Supplementary Fig. 8a). If VIP neurons constituted a dis-
crete functional cluster within the SCN's photoentrainment circuitry,
one would anticipate a higher in-group correlation coefficient for VIP+
neurons. Yet, our results demonstrated that in-group correlations for
VIP+ neurons did not significantly deviate from correlation coefficients
between VIP+ and VIP- neurons across all examined time points (Fig. 4d
and Supplementary Fig. 8b). Intriguingly, only during the photo-
entrainment dead zone at ZT 8 did VIP+ neurons exhibit a notably
higher correlation than VIP- neurons, suggesting a potential distinct
role during this specific phase (Fig. 4d, right panel). Further

exploration into the contributions of VIP neurons within the dead zone
of photoentrainment is required to decipher this phenomenon. Col-
lectively, these findings challenge the canonical view of a distinct VIP
neuron-led functional hierarchy for photoentrainment, highlighting a
surprising complexity among neuronal light responses within the SCN.

**Time-Dependent Variation in SCN Neuronal Responses to Light**
Circadian photoentrainment is characterized by three discrete light
responses. Here we analyzed the recording of 113 neurons to dissect
the light response dynamics of SCN neurons at a single-cell level across

these temporal zones. The overall averaged raw fluorescence intensity from the whole GRIN lens field of view is higher at ZT 8 (Supplementary Fig. 9a), similar to previous reports. Our comparative analysis of seven response clusters at different ZTs revealed no significant differences in most clusters at the population level (Supplementary Fig. 9b). Notably, the proportion of neurons exhibiting an inhibitory response (clusters 7 or combined cluster 6 with 7) was significantly reduced at ZT 8 (Fig. 5a), reflecting the subtle differences in population-wide light response patterns (Fig. 3a and Supplementary Fig 7). When the heat-map was organized in the same order as ZT 16, stark contrasts in individual neuronal responses emerged across the three-time points (Fig. 5b). To investigate the potential differences in SCN's time-gated network properties, we calculated the Pearson correlation coefficient of fluorescence intensities between neurons across all recorded data-sets. If SCN is comprised of a stable, time-independent network, one would expect to observe consistent light responses across different ZTs for individual neurons, which will generate a high correlation coefficient. Surprisingly, our cross-time correlation analysis during the

90-second light exposure revealed correlation coefficients close to zero (R-cross, ZT 8 – ZT 16, ZT 16 – ZT 22, and ZT8 – ZT 22, etc.) (Fig. 5c blue outline, Fig. 5d). Additionally, the correlation coefficients for different neurons at the same ZT time within a single trial (R-same, ZT 8, ZT 16, ZT 22) (Fig. 5c yellow outline) were significantly higher than the R-Cross values (Fig. 5d). These findings indicate that SCN neurons show some degree of synchronization within each trial. However, light responses for individual neurons vary with different trials and ZT, suggesting a dynamic functional network governing circadian photoentrainment.

To determine whether this network is driven by local connections, we identified neurons with high connectivity based on their correlation coefficients (r > 0.5). We found three distinct functional connectivity maps corresponding to ZT 8, ZT 16, and ZT 22, with no clear spatial clustering among the highly connected neurons (Fig. 5e and Supplementary Fig. 10). We then mapped the correlation coefficients against the inter-neuronal distances during the light exposure period across all three time points. A locally driven network would typically exhibit a

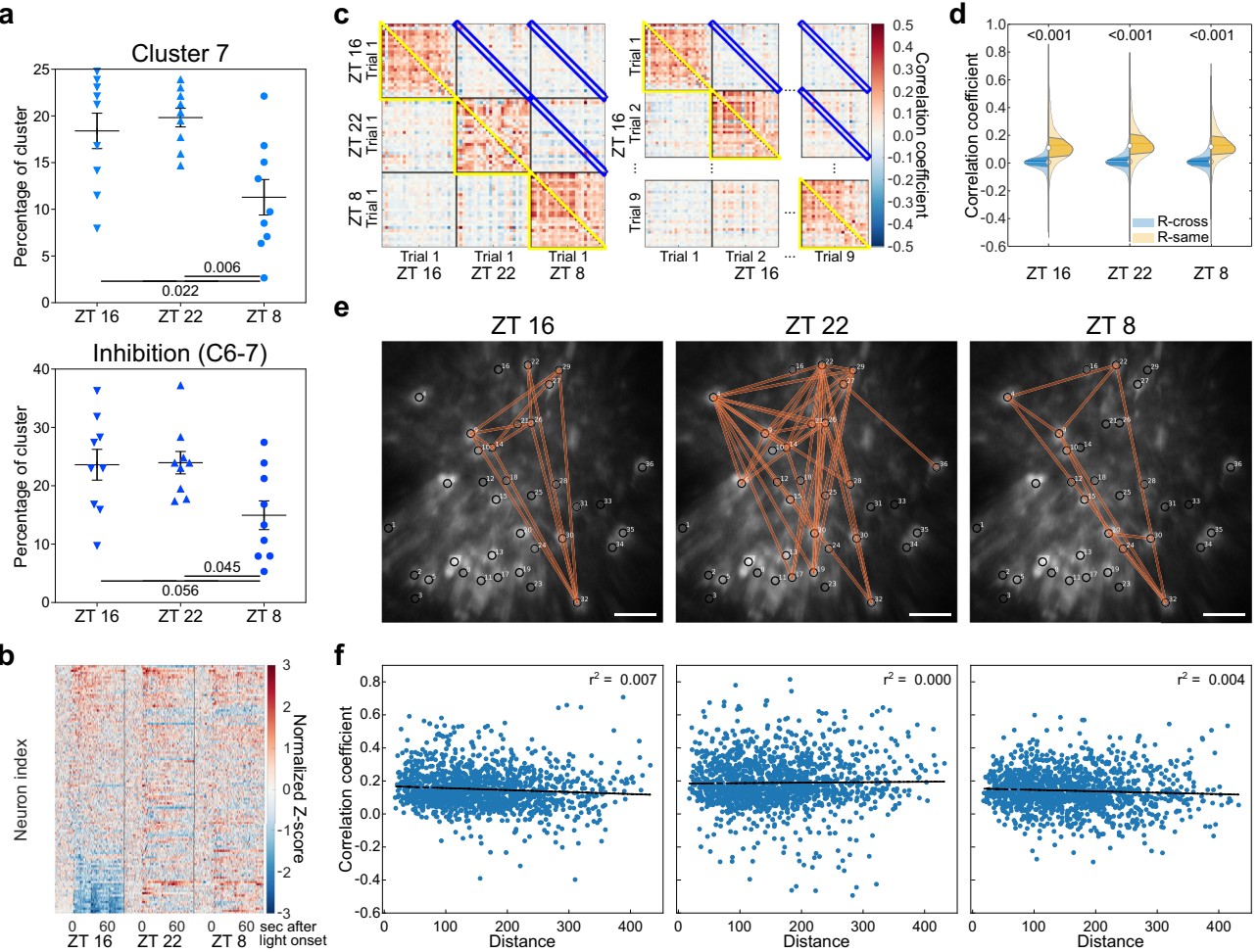

**Fig. 5 | Dynamic light responses from individual SCN neurons that form distinct temporal circuit. a** Percentage of inhibitory clusters (Cluster 7 and C6 - 7) is significantly decreased at ZT 8. Error bars indicate mean with SEM. **b.** Heat map of combined normalized traces from three-time points for 113 identified neurons ranked by mean Z-score at ZT16. **c** Representative heat maps of Pearson correlation coefficients comparing light responses between neurons. Left panel shows comparison between different ZTs, right panel shows comparison between different trials within the same ZT. Blue outlines indicate same neurons in different trials, and yellow outlines indicate different neurons in same trials. **d** Cross-time correlation analysis from each neuron between different trials (R-cross, blue outline in c and blue plots in **d**) is significantly lower than the correlation coefficients for different

neurons at the same ZT time within a single trial (R-same, yellow outline in c and orange plots in **d**). **e.** Representative diagram showing highly correlated neuron pairs (average r from 9 repeats > 0.5) from each time points. **f.** Scatter plots of distance between pairs of neurons and their Pearson correlation coefficient in respective time points. The r² value of linear regression lines is 0.007, 0, and 0.004 for ZT 16, ZT 22, and ZT 8, respectively. For (**a**) and (**d**), numbers indicate p-value with one-way ANOVA, Tukey post hoc tests. Scale bars are 100 μm, according to the imaging side of the GRIN endoscope. n = 3 animals. For violin plots, circle indicate median, thick vertical lines indicate interquartile ranges and thin vertical lines indicate 1.5X interquartile range.

negative correlation between neuronal distance and correlation coefficient. Contrary to this expectation, the relationship between the neurons' connectivity and their physical proximity was negligible, with an average r-squared value of only 0.04 (Fig. 5f). Collectively, these observations suggest that SCN neuronal light responses are orchestrated by a complex network architecture in 3-dimensional space rather than simple local circuits.

### Identification of Three Distinct SCN Neuronal Groups and Their Roles in Phase Shifting and Circadian Computation

To determine if the SCN consists of neuron populations with time-dependent light responses, we conducted additional analyses combining light responses from three distinct time points of each neuron. First, we calculated the average fluorescence z-score for individual neurons in five-second bins to reduce the effect of transient calcium waves. Subsequent ANOVA tests compared 18 light exposure bins to baseline across nine trials for each neuron. Bins showing significant

differences in post-hoc tests (Fig. 6a) were marked in orange to indicate significantly higher fluorescence intensity, and in blue for significantly lower intensity compared to their respective baselines (Fig. 6b). While most neurons exhibited varying light responses in each trial, only a small population showed consistently significant positive or negative responses. Through k-means clustering of data from all 54 bins across the three-time points, we identified three distinct groups of neurons (Fig. 6b). Specifically, one group displayed a persistent positive light response at ZT 16 (group 1, 11.5%), another showed consistent inhibition only at ZT 22 (group 2, 4.4%), and the majority did not exhibit a consistent light response (group 3, 84.1%), resulting in an overall flat response curve (Fig. 6c). To further confirm that these three groups of neurons are different from each other, we performed area under curve (AUC) calculation of their fluorescence response using normalized Z-score between 50-75 sec when light response were stabled (Fig. 6c). Statistical analysis confirmed that group 1 neurons showed significantly higher AUC compared to group 2 and 3 at ZT 16,

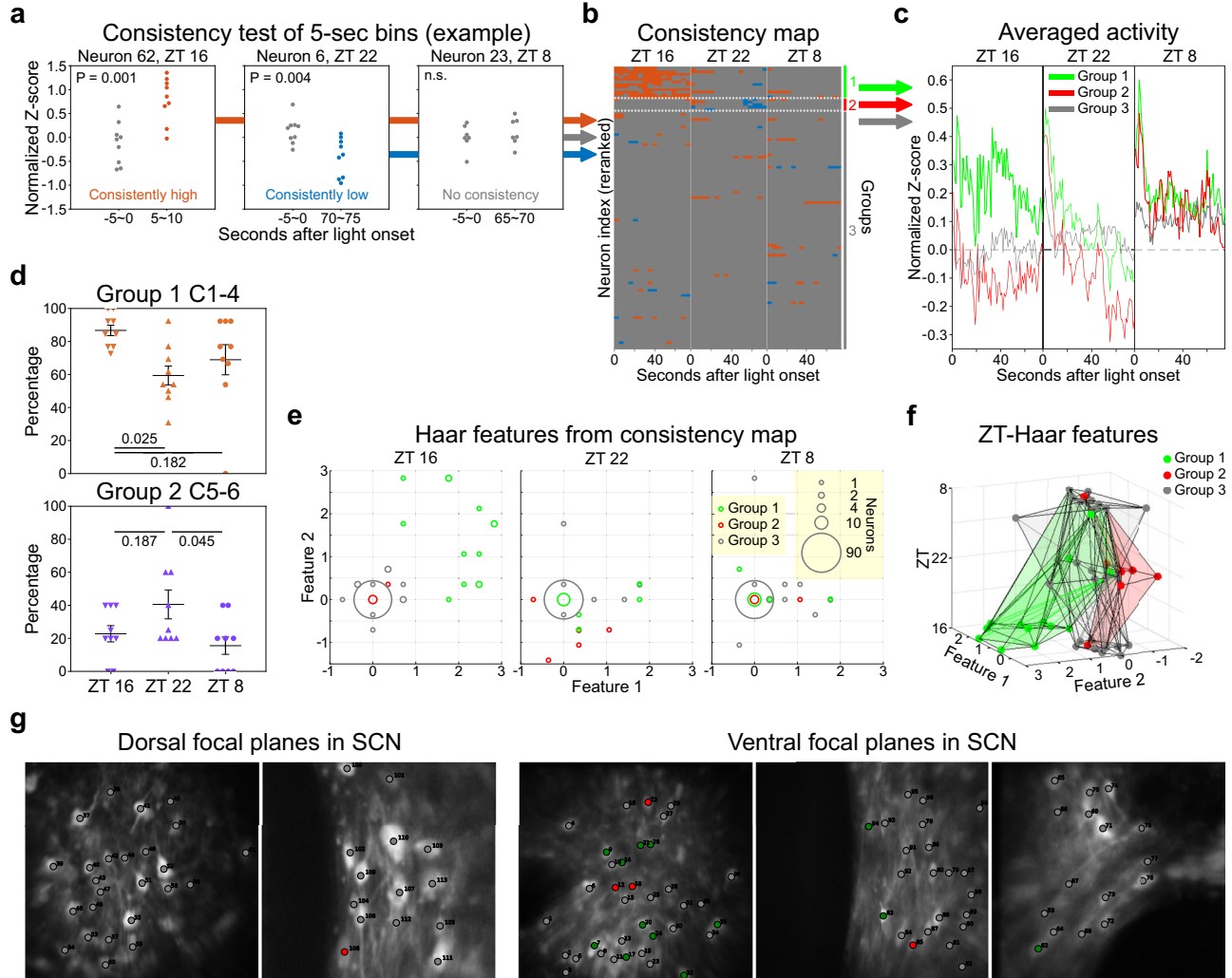

**Fig. 6 | Three distinct groups of SCN neurons with different time-dependent light responses. a** Representative light response analysis from 3 neurons. Three possible results are demonstrated here, including that the Z-score of the specific time point is significantly higher (orange), lower (blue), or not significantly changed (grey) compared to the baseline (one-way ANOVA and Tukey test). **b** The results in (**a**) are plotted together. Neurons are classified into three groups according to their significance pattern across all time points, separated by dotted lines. Group 1 neurons (green) are specific activated at ZT 16. Group 2 neurons (red) are specifically inhibited at ZT 22. While group 3 neurons (grey) do not show consistent positive or negative light response. **c** Averaged light-response traces from three

groups of neurons at different time points. **d** Group 1 neurons show significantly higher activation clusters (C1 - 4) at ZT 16 and group 2 neurons show significantly higher inhibition clusters (C5 - 6) at ZT 22. Numbers indicate p-values from one-way ANOVA, Tukey post hoc tests. Error bars indicate mean with SEM. **e** Haar feature analysis of all neurons for each ZT using 18 bins of 5 sec average fluorescence intensity. **f** Neuron trajectories in the Haar-ZT space show distinct characters from three groups. **g** Neuron groups are labeled on the raw images. Green dots indicate the cell body of group 1 neurons, red dots indicate group 2 neurons, while grey dots indicate group 3 neurons. Scale bars are 100 μm, according to the imaging side of the GRIN endoscope. $n = 3$ animals.

while group 2 neurons showed significantly lower AUC compared to group 3 at ZT 22. (Supplementary Fig. 11). When we compare the light response composition, group 1 neurons showed a higher percentage of activation clusters (cluster 1-4) at ZT 16, and group 2 neurons showed a higher percentage of inhibition clusters (cluster 5 + 6) at ZT 22 (Fig. 6d). Therefore, light responses from 3 groups of neurons can be separated into at least three classes statistically. Next, we applied the Haar transform, a dimensionality reduction technique for digitized data series composed of -1, 0, and 1, to each neuron in order to retain the dominant trends in the data while reducing the influence of fluctuations caused by noise. As with most dimensionality reduction methods, the spatial distribution of the Haar features reflects the similarities within the original datasets. By analyzing the polarity of 18 5-second bins from each ZT, our result revealed distinct patterns for each group in the Haar-ZT space. Interestingly, group 1 was differentiated from groups 2 and 3 at ZT16, whereas group 2 was distinct from groups 1 and 3 at ZT22. No clear segregation was observed among the three groups at ZT8 (Figs. 6e, f). Interestingly, both VIP+ and VIP- neurons were present in groups 1 and 2, with neurons in group 1 exclusively observed in the ventral focus planes of our recordings (Fig. 6g). It is noteworthy that most group 3 neurons (72.6%) still displayed a range of light responses throughout the nine trials at specific time points. The diversity of their light response types was significantly higher than that observed in baseline (Supplementary Fig. 12). Collectively, these results demonstrate that the SCN's functional network operates on a time-gated mechanism. Moreover, we have identified two subpopulations of SCN neurons with consistent light responses at ZT 16 or ZT 22, respectively, while the majority show dynamic light responses between trials. Together with behavioral results, our findings indicate the presence of a bi-stable circuit that can lead to at least two distinct time-gated functional networks within the SCN for circadian photoentrainment.

## Discussion

In this paper, we demonstrate the circadian time-dependent heterogeneity of SCN neurons in response to acute light, indicating a dynamic network within the SCN for circadian phase responses. Despite decades of study, several physical barriers have hindered the direct observation of the SCN in living animals. Located deep within the brain and hypothalamus, nearly 6 mm from the skull surface in mice, the SCN's accessibility to most optical research approaches is obstructed. Additionally, the SCN, being a small nucleus of 400 micrometers or less in every dimension and consisting of approximately 20,000 neurons, limits the feasibility of electrical recording. Consequently, ex vivo SCN preparations on brain slides have predominated as the methodology for studying mammals' central clock. However, this approach severs the SCN's endogenous network from RHT input and other brain regions, potentially altering the master clock's innate properties, including photoentrainment functions. Through the use of in vivo calcium imaging combined with a two-photon microscope and GRIN endoscopes, we have identified several emergent properties of the SCN. These include immediate responses to acute light stimulation, circadian time-dependent variable light response patterns, and neurons that are selectively responsive to light at specific circadian times. These emergent properties highlight the necessity of an in vivo setup with minimal network interruption to understand the central clock's computational network, which shall also be beneficial for studying other brain nuclei.

In our study, we found that light pulses during the day induced a significantly lower percentage of inhibitory neurons compared to nighttime. However, unlike previous studies that showed a higher percentage of activation at night than during the day using single-unit electrophysiological recording[63], we did not observe a significantly higher activation ratio during the daytime. Notably, their recording sites were primarily located in the dorsal part of the SCN. Our

recordings contain VIP neurons, and since VIP neurons are located in the ventral to middle SCN, this suggests that our recording location is likely in the ventral to the middle part of the SCN. Therefore, the ventral and dorsal parts of the SCN may generate different light responses at distinct circadian times. A novel method to record the entire dorsal-ventral axis of the SCN in vivo will be required to test this hypothesis. In addition, Meijer et al.'s in vivo multi-unit recordings also showed that light-induced multi-unit activity is higher at night than during the day[64]. However, the largest difference between day and night was observed under conditions of approximately one lux. Furthermore, the light response difference between day and night decreased as light stimulation became stronger. Since we used 722 lux blue light, our light intensity was much higher than in their recordings. Combining both results, it is possible that rod and cone signals were the primary factors generating higher SCN activity at night under low light conditions. However, under high light intensity, melanopsin-dependent signals may produce a consistent activation ratio in SCN neurons, regardless of the circadian time. For in vivo SCN calcium imaging, our raw traces of light responses were strikingly similar to those reported by Kahan et al.[65]. We demonstrated that most SCN neurons are light-responsive and consist of seven distinct clusters, including multiple inhibitory populations—more than the three clusters reported by Kahan et al. However, unlike Jones et al., who reported a significantly lower light response during the daytime in VIP neurons using fiber photometry[36], we did not observe such a decrease (Fig. 4). Unfortunately, using the two-photon GRIN lens recording system, we found it challenging to compare fluorescence intensity across different trials without a consistent marker for normalization. Furthermore, we recorded only 46 VIP neurons from three mice, which may suggest that our sample size was too small to replicate the results observed with fiber photometry. Nevertheless, we observed that the first peak after light exposure at ZT 8 had a longer peak time compared to ZT 16 and ZT 22 (Fig. 4a), indicating an overall weaker response. Together, our results showed some consistency with previous literature and provided additional insights into the complex neuronal circuitry of the SCN that conveys light information for circadian photoentrainment.

Our findings suggest an unconventional information flow in the SCN to process ambient light information. The traditional circuit was described as a simple linear circuit from the RHT to the VIP and GRP neurons-enriched ventrolateral "core", and then to the AVP neurons-enriched dorsomedial "shell" region. In this classic view, the PRC should be driven by a specific subset of neurons in SCN that are constantly responsive to light input. In other words, a certain subset of neurons in the SCN would show consistent light responses in all three circadian time zones. However, our findings show that distinct neuron groups are responsive to light at different circadian time zones respectively. Combined with recent anatomical studies showing dense ipRGC axonal fibers in the SCN[17–19,66], our results indicate that a complex network structure including many feedback circuits in SCN may better describe circadian photoentrainment than the conventional simple hierarchical model.

In our study, we have shown that neurons responsive to acute light in the SCN can be categorized by their light response. We identified distinct "outcome" neurons (group 1 and group 2 in Fig. 6) that respond specifically to light during the time zones associated with phase delays or advances. This suggests that the bi-directionality of the PRC consists of at least two distinct functional circuits within the SCN: neurons activated during early subjective night that contribute to phase delays, and neurons inhibited during late subjective night that contribute to phase advances. Our findings are in line with previous research identifying a spatiotemporal light-responsive pattern in the SCN[30,40]. Behaviorally, we demonstrated that activation of CT 16-trapped neurons leads to phase delays across all time zones, supporting the idea of separable delay and advance functional circuitries within the SCN. Prior research indicated a partial PRC by activating VIP

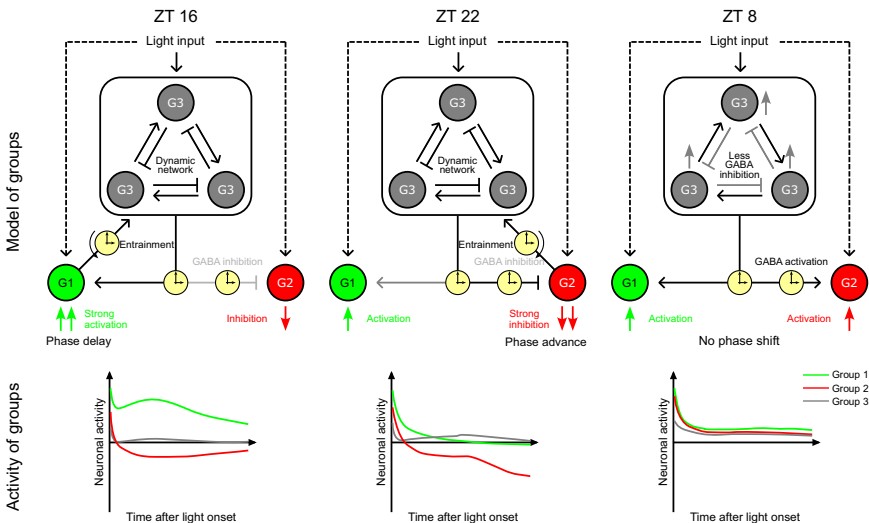

**Fig. 7 | Hypothetic model of SCN circuit for circadian photoentrainment.** Proposed bi-stable functional circuit within the SCN comprised with 3 distinct groups of neurons according to Fig. 6b (classification), 6c (average light response) and Supplementary Fig. 12 (dynamic response). Group 1 and 2 neurons are outcome units to drive phase shift at delay or advance time respectively. Group 3 neurons are the computational unit using populational dynamic coding. Right part: diagram of average group 1-3 neuronal response.

or CCK neurons[36,56], hinting that a subset of these may correspond to our identified delay or advance cells. Therefore, these neurons may be "outcome" neurons that contribute higher weight to synchronize the whole SCN population for phase shift. However, due to the absence of specific genetic tools for trapping inhibited neurons, we could not directly confirm the causal relationship between ZT 22 inhibition and the behavioral phase advances. Nevertheless, we could capture part of group 3 neurons defined in Fig. 6, which is the "computation" neurons using populational coding, in TRAP-CT 22 mice. Consequently, partial activation of these CT 22-trapped neurons produces a populational coding similar to light exposure. Therefore, activation of "computation" neurons does not solely induce phase advances but can also cause typical light-induced phase delays or no phase shift, depending on the time of activation. Moreover, the independent modulation of circadian phase shifts by two groups of neurons suggests a potential role in encoding day length and seasonal rhythms, which are not yet well understood in mammals.

Combining our results, we propose the existence of two 'time-gates' within the SCN's dynamic network. The first 'time-gate' operates similarly to a bi-stable transcriptional network model seen in cell fate differentiation. This gate could create two distinct low-entropy states within the SCN: one that leads to the activation of group 1 neurons, facilitating phase delay, and another that results in the inhibition of group 2 neurons, leading to phase advance. These low-entropy states reflect the dichotomous nature of the network's response to light, reminiscent of the robust and divergent outcomes observed in cellular differentiation pathways. The second proposed 'time-gate' in the SCN's dynamic network is characterized by a reduction in inhibition at ZT 8, potentially disrupting the feedback and feedforward interactions that are critical for the bi-stable model's function. This disturbance could prevent the network from settling into either of the two low-entropy states associated with phase shifts, explaining the absence of behavioral phase shifts after light exposure during the dead zone. An interesting analogy is the bi-stability in electronics. In a typical electronic bistable circuit, the basic components are feedback and feedforward between two inverters, which form a logic unit. Our findings of reduced inhibition during ZT 8 are also in line with previous research showing a diurnal switch in GABAergic signaling within the SCN, where it transitions from inhibitory to excitatory during the mid-day[67,68]. These results substantiate our proposed bi-stable network model and

support the validity of our observations and theoretical framework for circadian photoentrainment in the SCN (Fig. 7). According to Fig. 6, we can define two sets of outcome neurons as group 1 and group 2. They are either activated or inhibited specifically at ZT 16 or ZT 22. The remaining dynamic light-responsive neurons are classified as circadian computational group 3. Light input at a specific circadian time will activate different circuit for corresponding phase shift response.

Surprisingly, we also found that most neurons exhibited dynamic light response types across trials. The observed stochastic response of neurons to light within the SCN population, despite the seemingly straightforward process of phase shifting, suggests a highly flexible network. Recent studies have proposed a dynamic population coding in the cortex[69]. Here we observed a similar feature in the SCN circuit. Therefore, this dynamics network property may be a universal feature of the mammalian central nervous system that is integrated into the SCN for photoentrainment computation. This flexibility allows for a large pool of neurons to be available for input and computation, with subsets being utilized in a seemingly random fashion for each trial. This strategy preserves the system's plasticity for the learning process and offers redundancy to safeguard against damage. Such a mechanism ensures that even if individual neurons are compromised, the network retains its computational capabilities, channeling the processing through a select group of outcome-determining neurons.

The methodology and analysis carried out in the current study also have their limitations. One limitation of our study is the slower temporal resolution of calcium imaging compared to electrophysiological recording techniques. While calcium imaging is effective for observing global light responses over relatively long timescales, it does not have the temporal resolution to resolve the difference between monosynaptic or polysynaptic responses, or detecting spikes in SCN neurons. Our technique primarily captures the overall light response of the SCN network, potentially obscuring more subtle, rapid synaptic events. Consequently, the similar responses we observed from VIP+ and VIP- neurons cannot be conclusively interpreted as evidence of similar synaptic connections from the ipRGCs. Furthermore, we could not differentiate whether the inhibition light response we observed is the result of di-synaptic input through SCN GABAergic signal or direct GABA signal from ipRGCs[70]. Another limitation arises from our focus on analyzing activity trends during 90 sec of light exposure, rather than the oscillation of calcium transients observed in

many SCN neurons. Previous research indicates that the frequency of spontaneous calcium transient events varies with the time of day in vivo[35]. By concentrating on the general light response across 27 trials, we have not characterized these oscillations due to their phase could not be aligned perfectly with our recording paradigm. Since the physiological function and origin of spontaneous calcium transient in the SCN neurons is still unclear, our approach may overlook the implications of calcium transient for understanding the complete picture of circadian photoentrainment and SCN functionality. Next, we recorded the calcium-mediated light responses in the SCN at two different Z positions, approximately 105 μm apart. Although the dorsal section, on average, contains fewer VIP⁺ neurons, we cannot conclusively determine whether the ventral or dorsal Z sections correspond to the core or shell regions of the SCN, respectively. It remains possible that both Z sections are situated within either the core or the shell region. The long GRIN endoscopes required for accessing and recording from SCN neurons possess high pitch and low numerical aperture (NA). This limitation restricts the excitation efficiency of GCaMP within the two-photon system, thereby constraining our ability to achieve clear recordings when attempting to span larger Z-axis distances between two focal planes. In the future, higher NA rod-shaped thin endoscopes with a diameter of 600 μm or less, which can be consistently inserted above the SCN, will be required to overcome this challenge. Finally, the phase shifts induced by CNO injection at CT 22 and CT 2 in TRAP-CT16 mice were relatively small compared to those observed at CT 16. There was no significant difference in phase shift between the control group and TRAP-CT16 mice when calculated using the acrophase. Therefore, we cannot entirely rule out the possibility that activating the CT 16 circuit generates a strong phase delay specifically at CT 16 but does not induce a phase shift at other time points. Nonetheless, this alternative explanation still supports our conclusion that the light-activated circuit in the SCN at CT 16 is distinct from that at CT 22.

In summary, our methodological approach, employing two-photon excitation for spatial analysis, offers a minimally perturbative tool for studying SCN networks longitudinally. This advancement opens doors to exploring the fine-grained spatial organization of the SCN at single cell level in vivo and its implications for circadian photoentrainment. The identification of distinct neuronal groups that are responsive to light at different circadian times provides new insights beyond the traditional hierarchical model of circadian photoentrainment. This discovery lays the groundwork for future research into the identification of molecular markers and functional roles of the SCN neuronal subtypes, illuminating the underlying mechanisms for matching the endogenous clock with the external light-dark cycle in mammals.

## Methods

### Animal

All experiments were performed using C57BL/6 mice (*Mus musculus*) raised under the 12:12 hr light-dark cycle at room temperature (24 °C) under the regulation of National Taiwan University IACUC. All mice were under ad libitum feeding conditions throughout the whole experiment. Mouse lines employed in the experiments included VIP-Cre (Jackson Labs, Vip^tm1(cre)Zjh, strain #031628), Ai14 mice (Jackson Labs, Gt(ROSA)26Sor^tm14(CAG-tdTomato)Hze, strain #007914) for in vivo imaging (VIP^Cre/+; Ai14^CAG-tdTomato/+), and homozygous and heterozygous (specified in Supplementary Fig. 3) Fos^2A-iCreER (TRAP2) mice (Jackson Labs, strain #030323) for behavioral tests. GRIN lens insertion surgeries and AAV injections were performed at the age of 3 (Fig. 6G ventral planes right), 7 (Fig. 6g ventral planes left), and 8 (Fig. 6g ventral planes middle) month old, while the in vivo calcium images were obtained at least 4 weeks after GRIN endoscope implantation. For TRAP experiment tests, AAV inject were performed at 2 months old, and wheel running tests were performed between 4 to 8 months old. Both male and female mice were included in this study.

### Stereotaxic Surgeries

All surgeries were performed with a stereotaxic platform and Hamilton syringes controlled by a motorized micromanipulator to ensure precise virus injection and lens implantation to the SCN. Mice were anesthetized by isoflurane inhalation (5% vapor for induction and 1·1.5% vapor for maintenance). For in vivo imaging, the GRIN endoscopes (0.6 mm in diameter and 7.4 mm in length, GRINTECH NEM-060-50-00-920-S-1.5 P) were implanted after 300 nL AAV9-GCaMP7f (Addgene #104488) injection. The injection and implantation trajectories for the GRIN endoscope were rolled 6° clockwise on the coronal plane. Injection needle tips and endoscopes were aimed at anterior-posterior (AP) -0.46 mm and medial-lateral (ML) -0.725 mm from the bregma, and DV -5.2 to -5.4 from the surface of the cerebral cortex. Endoscopes were protected by custom-designed head plates and all materials were secured using dental cement and superbond. For TRAP2 mice in the behavioral tests, 150 nL of AAV9-hSyn-DIO-rM3D(Gs)-mCherry or AAV9-hSyn-DIO-EGFP (Addgene #50458 and #50457) were bilaterally injected, aimed to AP -0.46 mm and ML ± 0.2 mm from the bregma and DV -5.76 from the surface of the cerebral cortex.

### Behavioral Phase Shift

To trap light-induced cFos-positive cells, the stock solution of 4-hydroxytamoxifen (4-OHT; Sigma, Cat# H6278) was prepared at 100 mg/mL in ethanol and stored at −80 °C for up to one month. Before usage, the stock solution of 4-OHT was mixed with sunflower seed oil (Sigma, Cat #s 259853) to the final concentration of 10 mg/mL. Immediately after a 10-min 900-lux light pulse at CT 16 or CT 22, TRAP2 mice were intraperitoneally (IP) injected with 4-OHT at 50 mg/kg dosage. After at least 2 days, mice were transferred to wheel-running cages and housed in LD with *ad libitum* food and water. After at least 7 days of free-running in constant darkness (DD), mice were IP injected with CNO at 1 mg/kg or saline at CT 2, CT 16, and CT 22 for chemogenetic manipulation. Clozapine-N-oxide (CNO, Sigma, Cat# C0832) solution was prepared at 0.2 mg/mL in saline. Phase shifts were determined by using activity onset analysis in Clocklab (Actimetrics, US). Phase shifts were assessed by calculating the time difference observed on a subsequent day after each drug injection to the regression lines. The days of drug administration were excluded from the analysis.

### In vivo Deep Brain Calcium Imaging

A custom-built two-photon microscope with a 23-Hz frame rate at 512×512 pixel resolution was employed to acquire calcium images. GCaMP7f and tdTomato were excited using a 920 nm pulse laser (Coherent, US) at 90 mW output measured after a Zeiss 20x/0.5 objective, whose numerical aperture matches that of the GRIN endoscope. Awake mice were mounted on a custom-designed two-dimensional tilting stage during the image acquisition sessions. A free-rotating wheel was set for mice to relieve stress from the physical constraints of their heads. Before each recording, mice were placed under the recording platform in darkness for 10 min for habituation. To stimulate retinal photosensitive cells, a 488 nm light diffused onto a 10 ×26 mm screen in front of the eyes of mice. The average power of light on the screen was 1.76 μW/mm², which is approximately 722 lux according to the CIE photopic luminosity function. GCaMP7f signal was detected using a 525/39 nm bandpass filter, and the tdTomato signal was detected using a 625/90 nm bandpass filter. Every recording session contains 3 trails for each focus plane and is at least 14 h apart from each other to avoid excessive photobleaching.

### In vivo Calcium Image Data Processing

To examine the dynamic responses of individual SCN neurons, we followed the outlined workflow[71]: Firstly, we corrected motion artifacts in the acquired image slices induced by mice movement[72]. Next, we

utilized Cellpose to segment cell regions, creating cell masks that enable the extraction of calcium traces from the acquired sequence of x-y image slices[73,74]. Furthermore, to eliminate the decrease in fluorescence intensity caused by photobleaching[66,67,75,76], we applied curve fitting to the time series data of each neuron, removing the decay trend. Finally, to make the fluorescent intensity of each neuron from all trials comparable, we standardized the intensity using the normalized Z-score method. Further details on each step are provided as follows.

Motion correction: As per the mechanics of our mouse imaging platform, motion during the imaging process can be regarded as horizontal movement devoid of shearing and rotation. Consequently, motion correction can be attained through translation transformation[77]:

$$\mathbf{u} = \mathbf{T}_{motion}\mathbf{w}, \tag{1}$$

where $\mathbf{w} = \left[w_x\ w_y\ 1\right]^T$ and $\mathbf{u} = \left[u_x\ u_y\ 1\right]^T$ are the space coordinates of a voxel before and after correction, respectively. $T_{motion}$ is the transformation matrix:

$$\mathbf{T}_{motion} = \begin{bmatrix} 1 & 0 & 0 \\ 0 & 1 & 0 \\ t_x & t_y & 0 \end{bmatrix}, \tag{2}$$

where the parameters $t_x$ and $t_y$ denote the displacement along the x-axis and y-axis of the image slices, respectively. These parameters can be obtained through optimization algorithms in conjunction with a similarity measure $C$ (Mattes mutual information metric):

$$\hat{\mathbf{T}}_{motion} = \operatorname{argmin}_{\mathbf{T}_{motion}} C\left(\mathbf{T}_{motion}, \mathbf{I}_{moving}, \mathbf{I}_{ref}\right), \tag{3}$$

Here, $\mathbf{I}_{ref}$ represents the reference image, and $\mathbf{I}_{moving}$ denotes the image requiring correction. The optimization algorithm employs a regular gradient descent method for iterative optimization. The image at the start of the sequence serves as the reference image. After estimating the matrix $\mathbf{T}_{motion}$ it is used to correct motion in the image $\mathbf{I}_{moving}$.

Cell segmentation and extraction of calcium traces: Following the removal of motion-induced artifacts, we calculated the average of all image slices and employed the resulting mean slice as input for cell segmentation, utilizing Cellpose. We use the pre-trained model "CP" provided by Cellpose, with the cell diameter parameter set to 25 pixels. Subsequently, the segmented cell regions were labeled. Considering potential slight variations in activated neurons across different experiments, we manually adjusted their indices to ensure the consistent identification of the same group of neurons. Finally, we computed the average trace of a specific neuron by utilizing the traces of the voxels within the region associated with that neuron.

Detrend: We utilized the trust region optimization method, which is a nonlinear least squares approach, to estimate the decay coefficient for each calcium trace. Subsequently, the decay trend was subtracted to produce a detrended calcium trace, which was then utilized for further analysis. Moreover, background intensity extracted from the out-of-endoscope pixels was also subtracted in this step to exclude those slight (<50 of intensity unit. Soma of neurons were mostly 4000-15000 units in comparison) yet detectable signal changes caused by stimulation light and other residual signal fluctuations in the acquisition system.

Normalization: To perform further analysis, we transformed the fluorescent intensity of each neuron from each trial using a Z-score-based method:

$$Z_{Normalized}(t) = \frac{F(t) - F_{baseline}}{F_{std}}, \tag{4}$$

where $Z_{Normalized}(t)$ represents the normalized Z-score of assigned time (frame), $F(t)$ represents the fluorescent intensity of assigned time after detrending, $F_{baseline}$ represents the median intensity of the baseline, and $F_{std}$ represents the standard deviation of intensity across the 120-second recording. Instead of using the classic Z-score that defines the average of all values as zero, we set the median of the 30-second baseline as zero to better illustrate and compute the light-evoked neuronal responses.

## Immunofluorescence

Mice were anesthetized with tribromoethanol (avertin, 250 mg/Kg) and perfused with 10 ml cold PBS followed by 40 ml 4% PFA for fixation. Brains were isolated and postfixed in 4% PFA for 24 h. Serial 80 μm coronal brain slices were obtained by vibratomes. Sections were blocked with 10% normal goat serum in PBS containing 0.2% Triton-X and incubated overnight at 4 °C with primary antibodies. Following the wash, sections were incubated for 2 h with fluorescently labeled secondary antibodies. Brain sections were stained with anti-cFos mouse monoclonal antibody (1:1000, Abcam, ab208942), anti-vasopressin rabbit antibody (1:1000, Immunostar, 20069), and anti-GFP chicken antibody (1:1000, Abcam, ab13970). Secondary antibodies used were goat anti-mouse IgG Alexa 488 (Invitrogen, A21121, 1:500), goat anti-rabbit IgG Alexa 633 (Biotium, 20123, 1:500), and goat anti-chicken IgG Alexa 633 (Invitrogen, A21103, 1:500). Sections were then mounted with DAPI-containing RapiClear (SUNJin lab, Taiwan) and visualized by a Leica confocal microscope (SP5). For immunostaining cell counting, we utilized a semi-automatic counter and object-based colocalization analysis (OBCA) tools from the ImageJ plugin[78].

## Statistics & Reproducibility

No statistical method was used to predetermine sample size. No data were excluded from the analyses. Animals were randomly assigned to each group.

## Reporting summary

Further information on research design is available in the Nature Portfolio Reporting Summary linked to this article.

## Data availability

This study includes no data deposited in external repositories. Source data are provided with this paper.

## Code availability

Code for motion correction used in this study was provided and can be downloaded at https://github.com/chenwunci/motion_correction_2D.

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

## Acknowledgements

This work was supported by the Taiwan National Science and Technology Council grant NTSC 112-2636-B-002-012, 112-2628-B-002-028, and 112-2321-B-002 -019 (to S.-K.C.) and National Taiwan University. We thank the Technology Commons, College of Life Science at National Taiwan University for technical assistance with confocal imaging and virus core for virus packing. We also thank Yin-Tzu Hsieh for generating the illustration in the figures, and all lab members from S.-K.C., S.-W.C. and S.-C.W. for their inputs and supports.

## Author contributions

Conceptualization, S.-K.C., S.-W.C. and S.-C.W.; methodology, P.-T.Y., K.-C.J., S.-K.C., S.-W.C., and S.-C.W.; experiment, P.-T.Y. and E.-P.C.; data analysis, P.-T.Y., K.-C.J., W.-C.C., and E.-P.C.; writing—original draft, P.-T.Y. and E.-P.C.; writing—review & editing, all; supervision, S.-K.C., S.-W.C., and S.-C.W.

## Competing interests

The authors declare no competing interests.
