## [Transparent Peer Review file · Nature Communications]

Discrete photoentrainment of mammalian central clock is regulated by bi-stable dynamic network in the suprachiasmatic nucleus

Corresponding Author: Professor Shih-Kuo Chen

Version 0:

Reviewer comments:

Reviewer #1

(Remarks to the Author)

In this manuscript by Yeh et al., the authors aim to understand how the brain's biological clock, the suprachiasmatic nucleus (SCN), encodes three components of circadian photoentrainment: phase delays during the early night, phase advances during the late night, and no shift during the dead zone. To test this, they use in vivo two photon microscopy to record calcium activity in SCN neurons in response to light pulses given at different times of day. They find that the SCN neural network's light responses are diverse and highly stochastic, but a small population is consistently activated by light at dusk and another population is inhibited by light at dawn. Finally, they use the TRAP system to specifically activate the population of SCN neurons that respond to light at dusk and find that activation of these neurons elicits phase delays at any time of day. There are many positive things about this manuscript, including the use of cutting-edge technology (two photon microscopy, Fos-TRAP) and the commendable extent of analysis performed. Additionally, the identification of multiple distinct light-responsive clusters in the SCN and the effect of time-trapped neurons on phase shifts is highly novel. However, the manuscript would be improved by addressing the following concerns.

Major concerns:

1. Organization and presentation.

The amount of information per figure is immense and the logical progression from figure to figure is not entirely clear. To me it seemed like we, as readers, were following the authors as they tried one analysis method then the next. I would recommend moving much of the analysis to the supplemental.

In addition, the "story" of the paper seems like it would benefit more from rearranging the figures. Specifically, if Figure 6 (the Fos-TRAP figure) was one of the first figures, then the remaining figures came afterwards, the intriguing results in Fig. 6 would provide a stronger impetus for performing the two-photon imaging to identify the light-responsive cells in Figs. 1-5. Finally, many of the graphs are hard to interpret (e.g. Fig. 5 "Haar features") and few or no details are offered in the figure legends or methods that help with clarification.

2. Data in context.

There is no real attempt in the discussion or elsewhere to explain discrepancies between their results and from prior research (e.g. Fig. 3 VIP vs. non-VIP). The authors argue that there are limited numbers of neurons that are activated by light at any time of day and that VIP/non-VIP neurons have no difference in their light responses, but this is counter to several studies using single- and multi-unit electrophysiology, bulk fiber photometry calcium imaging, and one-photon single-cell calcium imaging. For example:

Meijer et al. J Neurosci 1998, Fig. 6 shows multi-unit light responses of the SCN as a function of circadian time. There are clear time-of-day dependent responses to light that show the typical lack of response during the subjective day and large response during the subjective night.

Nakamura et al. 2004 Neurosci Letters, Table 1 shows the proportion of SCN single units that respond to light at different

times of day. They show that many more SCN neurons are activated by light during the night compared to light during the day (83 vs. 26%).

Jones et al. 2018 J Neurosci, Fig. 2 shows time of day-dependent bulk calcium responses of SCN VIP neurons to light that are highest during the subjective night and peak at subjective dusk.

Kahan et al. 2023 iScience, Figs. 5,6 show one-photon single cell calcium responses of SCN VIP neurons to light during the early night. They find that the majority of VIP neurons (74%) respond to light, with 43% of them responding strongly.

I do not at all doubt the results in the present study, but they need to be put into context with the literature. If the authors averaged the response profile of all the cells recorded over time, would they see comparable light response profiles to the Meijer and Nakamura studies? If not, what is it about the present study that leads to different results? Does specifically expressing GCaMP in SCN VIP neurons in the Jones and Kahan studies somehow differ from expressing pan-neuronal GCaMP in the SCN and identifying VIP neurons by their tdTomato expression? All of this needs to be put into context in the discussion.

Minor concerns:

1. Methods: The authors need to provide the full genotypes of the mice used and whether they were heterozygous or homozygous. Several studies have noted that homozygous VIP Cre mice are VIP mutants (e.g. Joye et al. 2020). If the authors are using homozygous VIP Cre mice, this would completely change the interpretation of their results. Also, anecdotally, the number of cells labeled in the Fos-TRAP mice varies if the mice are heterozygous versus homozygous.

Additionally, there is no rationale given for performing multiple trials of light presentation at each Zeitgeber time point. The benefit of doing these multiple trials is unclear (based on response kinetics of SCN neurons and, presumably, the calcium sensors, it is perhaps unsurprising that the exact same neurons do not respond identically from trial to trial). Also, the methods mention 6 trials per time point but the results mention 9 trials per time point.

2. Figs. 3, 4: The method of calculating the Pearson correlation between neurons is not explained. The authors are presumably comparing the z-score of fluorescence intensity over an 80 s window as in Fig. 3A but this approach would make more sense for a more sensitive technique such as electrophysiology. Given the amount of inherent noise in calcium imaging not related to neuronal activity (biological: calcium waves, intracellular calcium release, etc.; non-biological: photobleaching, changes in sensor expression, etc.), comparing point-to-point fluorescence will likely never yield a strong correlation even if you are looking at the same neuron over time. It might be more prudent to compare something like the number of fluorescent peaks, peak widths, peak heights, etc. to get a better understanding of this relationship – this is the difference between “are these neurons doing the exact same thing” vs. “do these neurons have similar patterns of activity,” which is needed to interpret the data provided.

3. Fig. 4: Fig. 4C is hard to interpret, especially the right panel. It seems like data was only plotted for trials 1, 2, and 9 but this is not clarified in the figure legend. The yellow and blue lines also need to be explained in the legend for this panel.

4. Fig 5. The graphs in Fig. 5A need to share the same X and Y axes and the Z score should be shown across the entire ~80 seconds for transparency. The Haar features in Fig. 5E and F are confusing and do not seem to add to the manuscript.

5. Fig. 6: The methods mention that CNO was injected at 5 time points but the data presented here and in the supplemental only show the results from three time points. Is there a reason for this?

The 4-OHT dose for the Fos-TRAP experiments (500 mg/kg) is extremely high. I have not seen any studies using higher than 50 mg/kg 4-OHT. If this is not a typo, this needs to be explained.

In Fig. 6C-E the representative actograms do not seem to represent the extent of the phase shifts depicted in the methods. By eye the only actograms that clearly represent the data as shown in Fig. 6F are CNO CT16, TRAP CT16; CNO CT16, TRAP CT22; and CNO CT22, TRAP CT22. The CNO CT16, TRAP CT22 results are extremely impressive and clearly depict what is shown in Fig. 6F. For the other panels, I would recommend selecting other actograms that better show the point the authors are trying to make. Additionally, I would suggest measuring phase shifts using an additional phase marker such as acrophase, which would be less influenced by day-to-day onset variability.

6. Typos:

Line 47: “consistent” should be “consistently”

Line 85: “does” should be “does not”

Line 106: “shift” should be “shifts”

Line 360: “be expected” should be “expect”

Line 368: “neuronal” should be “neurons”

Line 551: “spike” should be “spikes”

Line 564: Incomplete sentence, replace period with a comma and connect the following sentence.

Line 590: Add “the” after “with.”

Reviewer #2

(Remarks to the Author)

The study by the Chen lab explores the discrete photoentrainment of the mammalian central clock, with a particular focus on the suprachiasmatic nucleus (SCN). The use of in vivo two-photon microscopy with gradient-index (GRIN) endoscopes is a commendable approach for mapping neural activity in response to light. The identification of distinct light responses in SCN neurons, especially at ZT16, ZT22, and ZT6, is a valuable contribution. The use of the targeted recombination in active populations (TRAP) system for neuron labeling is an innovative method that facilitates the targeted manipulation of specific neuronal groups.

While the primary findings are intriguing, the data presented are largely descriptive and could benefit from further exploration into the underlying mechanisms. A deeper understanding of these mechanisms would enhance the study's contribution to the field. The paper has potential, but there are several areas that could be improved to increase its quality.

Major Comments:

The paper would benefit from a more detailed account of the experimental procedures. Specifically, it remains unclear whether the mice were placed on a free-rotating wheel under the two-photon microscope before the lights were turned off, and if so, the duration of their acclimation period prior to the experiments at ZT16 and ZT22 should be specified. Additionally, elucidating the feeding status of the mice would be crucial for replicating the study and for a comprehensive understanding of its results.

The intensity of the light used during the in vivo calcium imaging for light response experiments is not detailed. Given that a high light intensity of 900 lux was employed for behavioral experiments, it is essential to disclose whether similar or varying intensities were applied for the in vivo imaging to ensure the consistency of experimental conditions.

Considering the potential variability in responses to different light intensities, incorporating a range of light intensities in Figure 2 could reveal how stronger light pulses might recruit a larger number of SCN neurons or elicit more consistent light responses of SCN neurons.

Given the SCN's pronounced day-night intracellular Ca²⁺ rhythms, it is unexpected that the basal Ca²⁺ fluctuation was not quantified. This measurement could offer valuable insights into the SCN's natural oscillations and provide a reference point for assessing the effects of light pulses. It could also validate the effectiveness of the GRIN endoscopes for deep brain imaging.

A no light pulse control group for the TRAP experiments is necessary to account for the SCN's inherent day-night calcium rhythm. Without this control, it is challenging to discern whether the TRAP system is capturing neurons activated by the light pulse or those that are endogenously active during the ZT timepoint.

The paper introduces a dynamic network model with G1, G2, and G3 classifications for SCN neurons. However, the model appears to lack sufficient data to support these classifications. More empirical evidence and statistical validation are needed to strengthen this aspect of the study.

Minor Comments:

In the figures, "GCaMP" should be annotated with its full name, GCaMP7f, for clarity and consistency with scientific nomenclature.

There is no histogram in Extended Figure 6C. The inclusion of this histogram would be beneficial for a complete understanding of the data distribution.

The font used in Extended Figure 6 appears inconsistent with the rest of the document. Ensuring a uniform font style.

Reviewer #3

(Remarks to the Author)

In this paper by Yeh and colleagues, the authors report on a SCN-based, time-gated functional network that could be an underlying mechanism for the discrete properties of the PRC. This is an interesting concept and the experimental work is generally well-performed. I do however have some major and minor concerns that I would like to see addressed in full.

Major concerns.

All of the heroic cellular/genetic manipulation is, in my mind, not unequivocally linked to the stated clock output behavior at the organismal level.

To be clear, the cellular data is intriguing (I do like the premise that the authors are evolving), but their behavioral data is not at the same level, nor is it compellingly integrated with the cellular data.

Minor concerns.

The subject description doesn't mention sex, and ages were broadly described as 2-8 months of age.

Multiple mouse lines were used and I don't know if there is a PRC literature for these lines, I would want to confirm their PRC before talking about controlling phase.

Figure 6 is the only figure with behavioral data (~ 15 days DD, w/ 7 da before and 7 da after stimulus), and this is problematic.

The images are quite small and I don't see a marker for the exact time of the stimulus. Also, tau varies considerably, both between animals and pre- post- stimulus. The later can (does) create false interpretations of phase shifts.

The data summary panels (6F, 6G) show predominantly delays and fewer/smaller advances occurred. However, I don't see raw actogram advances (or transients), which is an omission in my mind.

Version 1:

Reviewer comments:

Reviewer #1

(Remarks to the Author)

I thank the authors for responding to many of my concerns. However, I still have a few issues prior to endorsing the manuscript for publication:

“Activating light-responsive neurons during the early subjective night (ZT 16) using the TRAP and DREADD systems can induce phase delays at any circadian time, effectively breaking the photoentrainment dead zone. In contrast, activating ZT 22-trapped light-responsive neurons produces phase shifts similar to those caused by light exposure.”

Since this is a major claim of the paper (“activation of CT 16-trapped neurons...breaks the circadian photoentrainment dead zone”), it is essential that we as readers believe the data as presented.

However, the TRAP-CT16, CT22 CNO and TRAP-CT16, CT2 CNO actograms in Fig. 1, the shifts are subtle and are largely obscured by the red and blue lines.

Additionally, in the TRAP-CT16, CT22 CNO (and, to a somewhat lesser extent, the TRAP-CT16, CT2 CNO) actograms depicted in the author's response to reviewers that show the best fit lines drawn over the acrophases, the shifts are extremely subtle and do not reflect the best fit lines drawn through the activity onsets in Fig. 1.

An alternative explanation for the data as presented is that TRAP-CT22 neurons exhibit a light-like phase response to CNO activation (delay when activated at CT 16, advance at CT 22, no change at CT 2). This is consistent with Fig. 1 in the manuscript where the phase marker is activity onset, as well as the figure provided in the response to reviewers where the phase marker is acrophase. However, TRAP-CT16 neurons only exhibit any response when activated by CNO at CT 16. This fits with the figure provided in the response to reviewers (essentially no acrophase phase shift at any time except for CT 16) and in the representative figures in Fig. 1.

A possible explanation for the discrepancy is how the authors calculate phase. As an example, in the representative figure in the response to reviewers for the TRAP-CT16, CT2-CNO, it looks like there may be a mild phase shift between the red line of best fit pre-CNO and blue line of best fit post-CNO). However, the authors fail to account for the phase advance that occurred on day 5 independent of experimental manipulation. The line of best fit post-CNO therefore looks to be delayed from the line of best fit pre-CNO. If the pre-CNO line of best fit was instead calculated from days 5-8, there would be no phase shift. There appears to be the same issue with the TRAP CT 16, CT2 CNO figure in Fig. 1D, where the pre-CNO onset line of best fit fails to account for the day 5 phase advance, giving a misleading post-CNO onset line of best fit.

I recommend that the authors double-plot their actograms and only draw the best fit lines on one of the plots, as in Grippo et al. 2017 Fig. 2, Jones et al. 2018 Fig. 5, Milosavljevic et al. 2016 Fig. 1, and other papers investigating phase shifts. This will allow the readers to make their best judgment about the extent of the phase shifts in response to CNO activation.

I also recommend the authors provide additional representative actograms in the supplemental data. I also still recommend the authors provide an additional reanalysis of the data the data using different phase markers. Even though the authors say the acrophases are unstable in their response to reviewers, if the onsets are not reliable either, it is hard to agree with the figure's conclusions.

Reviewer #2

(Remarks to the Author)

The authors addressed most of my questions, and the manuscript has improved as a result. However, I did not find a

response to my last major concern: 'The paper introduces a dynamic network model with G1, G2, and G3 classifications for SCN neurons. However, the model appears to lack sufficient data to support these classifications. More empirical evidence and statistical validation are needed to strengthen this aspect of the study.' Considering that the authors have already addressed my other major concerns, it is acceptable that this issue remains unanswered.

Version 2:

Reviewer comments:

Reviewer #1

(Remarks to the Author)

The authors have satisfactorily addressed my concerns. Thank you for your efforts in greatly improving this manuscript.

Reviewer #2

(Remarks to the Author)

The authors addressed my questions. The manuscript has been largely improved. I have no further questions.

Thank you for handling our manuscript and allowing us to revise the manuscript. All issues were addressed by performing new experiments or modifications in the manuscript. We have updated our manuscript according to the reviewer's comments with point-by-point responses listed below. We labeled specific parts that correlate to the reviewer's comments in blue in the revised manuscript. All figures are either modified or expanded. Four new extended figures are added. We appreciate both the editor and reviewers for their comments to elevate the quality of this manuscript.

Reviewer #1 (Remarks to the Author):

In this manuscript by Yeh et al., the authors aim to understand how the brain's biological clock, the suprachiasmatic nucleus (SCN), encodes three components of circadian photoentrainment: phase delays during the early night, phase advances during the late night, and no shift during the dead zone. To test this, they use in vivo two-photon microscopy to record calcium activity in SCN neurons in response to light pulses given at different times of day. They find that the SCN neural network's light responses are diverse and highly stochastic, but a small population is consistently activated by light at dusk and another population is inhibited by light at dawn. Finally, they use the TRAP system to specifically activate the population of SCN neurons that respond to light at dusk and find that activation of these neurons elicits phase delays at any time of day. There are many positive things about this manuscript, including the use of cutting-edge technology (two photon microscopy, Fos-TRAP) and the commendable extent of analysis performed. Additionally, the identification of multiple distinct light-responsive clusters in the SCN and the effect of time-trapped neurons on phase shifts is highly novel. However, the manuscript would be improved by addressing the following concerns.

Response: We thank the reviewer for their positive comment indicating our work is highly novel.

Major concerns:

1. Organization and presentation.

The amount of information per figure is immense and the logical progression from figure to figure is not entirely clear. To me it seemed like we, as readers, were following the authors as they tried one analysis method then the next. I would recommend moving much of the analysis to the supplemental.

Response: We thank the reviewer for critical feedback on the reader side. We

improved the description in the text to make our paper better comprehended by readers.

In addition, the “story” of the paper seems like it would benefit more from rearranging the figures. Specifically, if Figure 6 (the Fos-TRAP figure) was one of the first figures, then the remaining figures came afterwards, the intriguing results in Fig. 6 would provide a stronger impetus for performing the two-photon imaging to identify the light-responsive cells in Figs. 1-5.

Response: We thank the reviewer’s critical comment on the presentation of the story. We rearranged our paper following the suggestion by moving behavioral results to Figure 1.

Finally, many of the graphs are hard to interpret (e.g. Fig. 5 “Haar features”) and few or no details are offered in the figure legends or methods that help with clarification.

Response: We thank the reviewer for this critical comment. Additional descriptions of Haar features and other analyses were added to better illustrate the figures. For example, we added the description of the Haar feature in the text as follows:

“Next, we applied the Haar transform, a dimensionality reduction technique for digitized data series composed of -1, 0, and 1, to each neuron in order to retain the dominant trends in the data while reducing the influence of fluctuations caused by noise. As with most dimensionality reduction methods, the spatial distribution of the Haar features reflects the similarities within the original datasets”

2. Data in context.

There is no real attempt in the discussion or elsewhere to explain discrepancies between their results and from prior research (e.g. Fig. 3 VIP vs. non-VIP). The authors argue that there are limited numbers of neurons that are activated by light at any time of day and that VIP/non-VIP neurons have no difference in their light responses, but this is counter to several studies using single- and multi-unit electrophysiology, bulk fiber photometry calcium imaging, and one-photon single-cell calcium imaging. For example:

Meijer et al. J Neurosci 1998, Fig. 6 shows multi-unit light responses of the SCN as a function of circadian time. There are clear time-of-day dependent responses to light that show the typical lack of response during the subjective day and large response during the subjective night.

Nakamura et al. 2004 Neurosci Letters, Table 1 shows the proportion of SCN single units that respond to light at different times of day. They show that many more SCN neurons are activated by light during the night compared to light during the day (83 vs. 26%).

Jones et al. 2018 J Neurosci, Fig. 2 shows time of day-dependent bulk calcium responses of SCN VIP neurons to light that are highest during the subjective night and peak at subjective dusk.

Kahan et al. 2023 iScience, Figs. 5,6 show one-photon single cell calcium responses of SCN VIP neurons to light during the early night. They find that the majority of VIP neurons (74%) respond to light, with 43% of them responding strongly.

I do not at all doubt the results in the present study, but they need to be put into context with the literature. If the authors averaged the response profile of all the cells recorded over time, would they see comparable light response profiles to the Meijer and Nakamura studies? If not, what is it about the present study that leads to different results? Does specifically expressing GCaMP in SCN VIP neurons in the Jones and Kahan studies somehow differ from expressing pan-neuronal GCaMP in the SCN and identifying VIP neurons by their tdTomato expression? All of this needs to be put into context in the discussion.

Response: We thank the reviewer for helping us improve the quality of the current manuscript. We have updated the discussion section by adding comparisons to these references as follows:

“In our study, we found that light pulses during the day induced a significantly lower percentage of inhibitory neurons compared to nighttime. However, unlike previous studies that showed a higher percentage of activation at night than during the day using single-unit electrophysiological recording 73, we did not observe a significantly higher activation ratio during the daytime. Notably, their recording sites were primarily located in the dorsal part of the SCN. Our recordings contain VIP neurons, and since VIP neurons are located in the ventral to middle SCN, this suggests that our recording location is likely in the ventral to the middle part of the SCN. Therefore, the ventral and dorsal parts of the SCN may generate different light responses at distinct circadian times. A novel method to record the entire dorsal-ventral axis of the SCN in vivo will be required to test this hypothesis. In addition, Meijer et al.'s in vivo multi-unit recordings also showed that light-induced multi-unit activity is higher at night than during the day 74. However, the largest difference between day and night was observed under

conditions of approximately one lux. Furthermore, the light response difference between day and night decreased as light stimulation became stronger. Since we used 722 lux blue light, our light intensity was much higher than in their recordings. Combining both results, it is possible that rod and cone signals were the primary factors generating higher SCN activity at night under low light conditions. However, under high light intensity, melanopsin-dependent signals may produce a consistent activation ratio in SCN neurons, regardless of the circadian time. For in vivo SCN calcium imaging, our raw traces of light responses were strikingly similar to those reported by Kahan et al. 75. We demonstrated that most SCN neurons are light-responsive and consist of seven distinct clusters, including multiple inhibitory populations—more than the three clusters reported by Kahan et al. However, unlike Jones et al., who reported a significantly lower light response during the daytime in VIP neurons using fiber photometry 36, we did not observe such a decrease (Figure 4). Unfortunately, using the two-photon GRIN lens recording system, we found it challenging to compare fluorescence intensity across different trials without a consistent marker for normalization. Furthermore, we recorded only 46 VIP neurons from three mice, which may suggest that our sample size was too small to replicate the results observed with fiber photometry. Nevertheless, we observed that the first peak after light exposure at ZT 8 had a longer peak time compared to ZT 16 and ZT 22 (Figure 4A), indicating an overall weaker response. Together, our results showed some consistency with previous literature and provided additional insights into the complex neuronal circuitry of the SCN that conveys light information for circadian photoentrainment.”

Minor concerns:

1. Methods: The authors need to provide the full genotypes of the mice used and whether they were heterozygous or homozygous. Several studies have noted that homozygous VIP Cre mice are VIP mutants (e.g. Joye et al. 2020). If the authors are using homozygous VIP Cre mice, this would completely change the interpretation of their results. Also, anecdotally, the number of cells labeled in the Fos-TRAP mice varies if the mice are heterozygous versus homozygous.

Response: We thank the reviewer for pointing out the importance of heterozygous versus homozygous. We clarified the information in the “Animal” session in the “Material and Methods” part. In the current study, all mice for imaging were VIP^{Cre/+}; Ai14^{CAG-tdTomato/+} double heterozygous. For the Fos-TRAP experiment, we now removed heterozygous mice and only showed homozygous mice’s data. Although the n number is reduced, the result of the statistical analysis remains unchanged.

Additionally, there is no rationale given for performing multiple trials of light presentation at each Zeitgeber time point. The benefit of doing these multiple trials is unclear (based on the response kinetics of SCN neurons and, presumably, the calcium sensors, it is perhaps unsurprising that the exact same neurons do not respond identically from trial to trial). Also, the methods mention 6 trials per time point but the results mention 9 trials per time point.

Response: We thank the reviewer's point. There are two major reasons for us to do 9 trials of light response for each neuron. First, the fluorescence signal is very weak due to the low NA (0.5) of the GRIN lens. To overcome this issue, we perform 9 repeats to enhance the ability to do analysis. Eventually, we need to do PCA sorting to cluster each different type of light response. Second, the advantage of *in vivo* two-photon GRIN lens imaging is that we can observe the same neuron repeatedly. Therefore, to reduce the potential bias on single-day recording, we perform 3 sets of recording sessions on 3 different days. We added a description to strengthen the rationale for performing multiple trials and the experimental design to reduce confusion.

"Here, each recording session consisted of three trials of a 30-second dark baseline followed by a 90-second light exposure (488 nm, 1.76 $\mu\text{W}/\text{mm}^2$) to the eyes. Activity from two z-positions within the SCN, separated by 105 μm , was captured during the same session (Figure 2E). To enhance the statistical power, we repeated the same recording procedure for each ZT time from 3 different days to increase the total number of activity traces."

2. Figs. 3, 4: The method of calculating the Pearson correlation between neurons is not explained. The authors are presumably comparing the z-score of fluorescence intensity over an 80 s window as in Fig. 3A but this approach would make more sense for a more sensitive technique such as electrophysiology. Given the amount of inherent noise in calcium imaging not related to neuronal activity (biological: calcium waves, intracellular calcium release, etc.; non-biological: photobleaching, changes in sensor expression, etc.), comparing point-to-point fluorescence will likely never yield a strong correlation even if you are looking at the same neuron over time. It might be more prudent to compare something like the number of fluorescent peaks, peak widths, peak heights, etc. to get a better understanding of this relationship – this is the difference between “are these neurons doing the exact same thing” vs. “do these neurons have similar patterns of activity,” which is needed to interpret the data provided.

Response: We thank the reviewer for this critical comment. We also performed

waveform analysis such as peak-to-peak correlation analysis. Since calcium imaging in SCN neurons could not produce action potential-like signals similar to cortical principal neurons, the fluorescence intensity is already like performing temporal integration of neuronal activity. Therefore, comparing the overall trace could decipher the “do these neurons have similar patterns of activity”. As we predicted, using peaks of the calcium wave to do correlation analysis produced similar results to our original full 80-sec analysis. To make the analysis more comprehensive, we have added peak to peak correlation analysis in the new Extended Figure 7.

3. Fig. 4: Fig. 4C is hard to interpret, especially the right panel. It seems like data was only plotted for trials 1, 2, and 9 but this is not clarified in the figure legend. The yellow and blue lines also need to be explained in the legend for this panel.

Response: We thank the reviewer for pointing out the ambiguous description. We added more information in the figure legend to support those panels. Considering the limitation of the page, we only showed 3 out of 9 trials as representative data in Figure 5C. Here we provide the full heatmap of correlation analysis below. For one focus plan with 25 neurons, 9 trials of cross-correlation analysis generate a 225*225 tile heatmap. Since trial-to-trial comparisons are similar, we decided to crop the heatmap and only showed 3 trials of cross-correlation 75*75 tile heatmap. In the new figure, we added space between trial 2 and trial 9 to clarify that this is a cropped heatmap.

4. Fig 5. The graphs in Fig. 5A need to share the same X and Y axes and the Z score should be shown across the entire ~80 seconds for transparency. The Haar features in Fig. 5E and F are confusing and do not seem to add to the manuscript.

Response: We thank the reviewer for these comments. We adjusted the scales of the axes to make the figures more comparable. The entire ~80-second information was shown in Figure 5B (now 6B, after adjustment of the structure of our story) and the three panels in Figure 5A are examples to demonstrate how we label colors in Figure 5B. Additional descriptions of Haar features were added in the “Material and Method” and “Results” texts to better illustrate the figures.

5. Fig. 6: The methods mention that CNO was injected at 5 time points but the data presented here and in the supplemental only show the results from three time points. Is there a reason for this?

Response: We apologize for making this typo. We inject CNO at 3 time points. The method section has been updated.

The 4-OHT dose for the Fos-TRAP experiments (500 mg/kg) is extremely high. I have not seen any studies using higher than 50 mg/kg 4-OHT. If this is not a typo, this needs to be explained.

Response: We apologize for making this typo. It is indeed 50 mg/kg. We have updated the manuscript to show the correct dosage.

In Fig. 6C-E the representative actograms do not seem to represent the extent of the phase shifts depicted in the methods. By eye the only actograms that clearly represent the data as shown in Fig. 6F are CNO CT16, TRAP CT16; CNO CT16, TRAP CT22; and CNO CT22, TRAP CT22. The CNO CT16, TRAP CT22 results are extremely impressive and clearly depict what is shown in Fig. 6F. For the other panels, I would recommend selecting other actograms that better show the point the authors are trying to make. Additionally, I would suggest measuring phase shifts using an additional phase marker such as acrophase, which would be less influenced by day-to-day onset variability.

Response: We thank the reviewer for pointing out this comment. We have picked different representative actograms for the new updated Figure 1. However, the acrophase was relatively unstable in our case, perhaps due to stereotaxis surgery. Below we showed that acrophase from the same animal in Figure 1. D is zoomed in view for the bottom left actogram. Overall, the onsets are more stable than the acrophase. Therefore, we decided to use onset to calculate phase shift.

6. Typos:

Line 47: “consistent” should be “consistently”

Line 85: “does” should be “does not”

Line 106: “shift” should be “shifts”

Line 360: “be expected” should be “expect”

Line 368: “neuronal” should be “neurons”

Line 551: “spike” should be “spikes”

Line 564: Incomplete sentence, replace period with a comma and connect the following sentence.

Line 590: Add “the” after “with.”

Response: We thank the reviewer for kindly and carefully pointing out these typos. We corrected the typos followed the reviewer’s suggestions and checked grammar issues again.

Reviewer #2 (Remarks to the Author):

The study by the Chen lab explores the discrete photoentrainment of the mammalian central clock, with a particular focus on the suprachiasmatic nucleus (SCN). The use of in vivo two-photon microscopy with gradient-index (GRIN) endoscopes is a commendable approach for mapping neural activity in response to light. The identification of distinct light responses in SCN neurons, especially at ZT16, ZT22, and ZT6, is a valuable contribution. The use of the targeted recombination in active populations (TRAP) system for neuron labeling is an innovative method that facilitates the targeted manipulation of specific neuronal groups.

While the primary findings are intriguing, the data presented are largely descriptive and could benefit from further exploration into the underlying mechanisms. A deeper understanding of these mechanisms would enhance the study's contribution to the field. The paper has potential, but there are several areas that could be improved to increase its quality.

Response: We thank the reviewer for their positive comment indicating our work’s potential contribution to the field.

Major Comments:

The paper would benefit from a more detailed account of the experimental procedures. Specifically, it remains unclear whether the mice were placed on a free-rotating wheel under the two-photon microscope before the lights were turned off, and if so, the duration of their acclimation period prior to the experiments at ZT16 and ZT22 should be specified. Additionally, elucidating the feeding status of the mice would be crucial for replicating the study and for a comprehensive understanding of its results.

Response: We thank the reviewer for pointing out this missing information. We have

now added detailed information in the method section. Animals were placed on a free-rotating wheel under the two-photon microscope. They were habituated for at least 10 min before recording under darkness for all time points. Mice were all kept under ad libitum feeding.

The intensity of the light used during the in vivo calcium imaging for light response experiments is not detailed. Given that a high light intensity of 900 lux was employed for behavioral experiments, it is essential to disclose whether similar or varying intensities were applied for the in vivo imaging to ensure the consistency of experimental conditions

Response: We thank the reviewer for pointing out the unit conversion issue. We used LED light with a diffuser to provide light stimulation to the eye. The average power after the diffuser is $1.76 \mu\text{W}/\text{mm}^2$, which is approximately 722 lux according to the CIE photopic luminosity function. We have added this information to the method section.

Considering the potential variability in responses to different light intensities, incorporating a range of light intensities in Figure 2 could reveal how stronger light pulses might recruit a larger number of SCN neurons or elicit more consistent light responses of SCN neurons.

Response: We thank the reviewer for this comment. We indeed perform this experiment. However, the result is inconclusive. Therefore, before we can confidently draw a consistent conclusion, we decided to not include light-intensity results in the current manuscript. Furthermore, our light intensity is already high, which should adequately stimulate ipRGCs. Therefore, it is unlikely that the variability originates from the light intensity used in the current study.

Given the SCN's pronounced day-night intracellular Ca^{2+} rhythms, it is unexpected that the basal Ca^{2+} fluctuation was not quantified. This measurement could offer valuable insights into the SCN's natural oscillations and provide a reference point for assessing the effects of light pulses. It could also validate the effectiveness of the GRIN endoscopes for deep brain imaging.

Response: We thank the reviewer for this critical comment. Since the fluorescence intensity is very weak under the two-photon and GRIN lens system, we used Z-scores to analyze most of the recording traces. This method normalizes the intensity of all neurons relative to their respective baselines. As a result, we did not perform raw fluorescence intensity comparisons between different ZT in the original manuscript. Here, we compared the baseline GCaMP signal between different ZT using the raw

fluorescence intensity across the full field of view from all optical sections in 3 mice. We found that the average baseline fluorescence intensity at ZT 8 was significantly higher than at ZT 22. This result is consistent with previous *in vivo* photometry and *in vitro* recordings of the SCN in mice. We now added this comparison in Extended figure 8A.

A no-light pulse control group for the TRAP experiments is necessary to account for the SCN's inherent day-night calcium rhythm. Without this control, it is challenging to discern whether the TRAP system is capturing neurons activated by the light pulse or those that are endogenously active during the ZT timepoint.

Response: We thank the reviewer for this specific comment. Although previous literature showed that cFos expression in the SCN displays daily oscillation, the nighttime cFos is much lower than daytime. Therefore, the TRAP system should primarily label light-induced cFos neurons in the SCN at CT 16 and CT 22. To confirm this within the 3-month revision period, we conducted 1 experiment to TRAP CT 16 SCN neurons under dark conditions and inject CNO at CT 16 to test their phase shift response. We found that labeled neurons in the SCN from dark-TRAP-CT 16 mice (new data) are significantly lower than regular TRAP-CT 16 mice (original data). Furthermore, CNO injection at CT 16 showed a significantly lower phase shift in dark-TRAP-CT 16 mice compared to regular TRAP-CT 16 mice. The phase shift in dark-TRAP-CT 16 mice after CNO injection is not significantly different from zero (new Extended Figure 4). Together, it confirms that the original behavior results primarily originated by trapping light-responsive neurons under CT 16. Unfortunately, due to the limit of time, we could not perform the whole set of experiments similar to Figure 1. However, our single time point test does strongly argue against the potential influence of endogenous cFos-positive neurons during the dark phase.

The paper introduces a dynamic network model with G1, G2, and G3 classifications for SCN neurons. However, the model appears to lack sufficient data to support these classifications. More empirical evidence and statistical validation are needed to strengthen this aspect of the study.

Minor Comments:

In the figures, "GCaMP" should be annotated with its full name, GCaMP7f, for clarity and consistency with scientific nomenclature.

Response: We thank the reviewer's comment and the texts in the figures were adjusted accordingly.

There is no histogram in Extended Figure 6C. The inclusion of this histogram would be beneficial for a complete understanding of the data distribution.

Response: We apologize for this error. One bar in Extended Figure 6C is missing in the PDF merging process. We have updated the figure and it has been moved to Extended Figure 1.

The font used in Extended Figure 6 appears inconsistent with the rest of the document. Ensuring a uniform font style.

Response: We thank the reviewer for pointing out the inconsistency issues. We have corrected those issues.

Reviewer #3 (Remarks to the Author):

In this paper by Yeh and colleagues, the authors report on a SCN-based, time-gated functional network that could be an underlying mechanism for the discrete properties of the PRC. This is an interesting concept and the experimental work is generally well-performed. I do however have some major and minor concerns that I would like to see addressed in full.

Response: We thank the reviewer for their positive comment indicating our work is well-performed.

Major concerns.

All of the heroic cellular/genetic manipulation is, in my mind, not unequivocally linked to the stated clock output behavior at the organismal level.

To be clear, the cellular data is intriguing (I do like the premise that the authors are evolving), but their behavioral data is not at the same level, nor is it compelling integrated with the cellular data.

Response: We have rearranged the presentation sequence for the manuscript to increase readability. Our behavior data strongly indicate at least 2 distinct time-dependent function circuits in the SCN for phase delay and phase advance in mice. Our in vivo functional imaging results also showed that at least 3 groups of neurons showed distinct temporal light responses. Therefore, the results from both behavioral and functional studies are correlated and suggest a bi-stable functional circuitry for SCN to produce circadian photoentrainment. The functional study also showed dynamic light response for individual SCN neurons. This dynamic response could also explain why

TRAP-CT 22 mice showed 3 different phase responses at CT 2, CT 16, and CT 22. Therefore, we strongly believe our behavioral and functional data are in support of each other.

Minor concerns.

The subject description doesn't mention sex, and ages were broadly described as 2-8 months of age.

Response: We thank the reviewer for pointing out this issue. We have now added specific details in the Methods section. In this study, we used both males and females to avoid bias towards a specific sex. The broad age range was due to the extended duration of the experimental design. We have now included all specific details in the Methods section as follows:

“GRIN lens insertion surgeries and AAV injections were performed at the age of 3 (Figure 6G ventral planes right), 7 (Figure 6G ventral planes left), and 8 (Figure 6G ventral planes middle) month old, while the in vivo calcium images were obtained at least 4 weeks after GRIN endoscope implantation. For TRAP experiment tests, AAV inject were performed at 2 months old, and wheel running tests were performed between 4 to 8 months old. Both male and female mice were included in this study.”

Multiple mouse lines were used and I don't know if there is a PRC literature for these lines, I would want to confirm their PRC before talking about controlling phase.

Response: We thank the reviewer for the critical comment. All of our experimental mice were under the C57BL/6 stain background. It has been used in many photoentrainment studies worldwide. The specific surgery and TRAP mice should not significantly change the phase advance, phase delay, and dead zone response to light. We did perform light pulse experiments from the same set of mice at CT 8, CT 16, and CT 22. We did not include these results originally because we believe they were redundant. Now we add them as new Extended Figure 2 to show they have normal light-induced phase shift response.

Figure 6 is the only figure with behavioral data (~ 15 days DD, w/ 7 da before and 7 da after stimulus), and this is problematic.

Response: This behavioral data was very challenging. TRAP mice need to undergo precise stereotaxis surgery to inject AAV-DREAAD in the SCN. After recovery, careful handling is required to TRAP light-responsive neurons at specific circadian times. Each

mouse was tested with light pulses at 3 different time points for regular light-induced phase shift, CNO-injected phase shift, and control PBS-induced phase shift. The whole set of experiments requires 5 months to complete. We also confirm the injection site post hoc after all behavior tests. Data from animals without clear injection in the SCN were discarded. Furthermore, our in vivo recording also suggested different subpopulations of neurons in the SCN. Both behavioral and functional imaging data indicate that there are multiple functional circuits in the SCN for circadian photoentrainment. We strongly believe our behavioral data are highly significant.

The images are quite small and I don't see a marker for the exact time of the stimulus. Also, tau varies considerably, both between animals and pre- post- stimulus. The later can (does) create false interpretations of phase shifts.

Response: We thank the reviewer for this suggestion. We have changed the marker for CNO or PBS injection on the figure and made them larger to increase readability. The tau varies may be the result of stereotaxis surgery to inject AAV vector in the SCN. However, since control mice were also under similar procedures. This confounding factor has been controlled in our experimental design. Here we provide a tau analysis before and after CNO injection. Although there is a slight difference between each experiment, populational-wise there is no significant change in tau after CNO injection.

The data summary panels (6F, 6G) show predominantly delays and fewer/smaller advances occurred. However, I don't see raw actogram advances (or transients), which is an omission in my mind.

Response: We thank the reviewer for this critical comment. We randomly selected actograms in our original figure. To reduce confusion, now we have updated the figure to show better representative actograms for our statistical result.

Reviewer #1 (Remarks to the Author):

I thank the authors for responding to many of my concerns. However, I still have a few issues prior to endorsing the manuscript for publication:

“Activating light-responsive neurons during the early subjective night (ZT 16) using the TRAP and DREADD systems can induce phase delays at any circadian time, effectively breaking the photoentrainment dead zone. In contrast, activating ZT 22-trapped light-responsive neurons produces phase shifts similar to those caused by light exposure.” Since this is a major claim of the paper (“activation of CT 16-trapped neurons...breaks the circadian photoentrainment dead zone”), it is essential that we as readers believe the data as presented.

We thank the reviewer for agreeing with most of our previous responses.

However, the TRAP-CT16, CT22 CNO and TRAP-CT16, CT2 CNO actograms in Fig. 1, the shifts are subtle and are largely obscured by the red and blue lines.

We thank the reviewer for this important issue. Indeed, the phase shift caused by CNO injection at CT 22 and CT 2 in TRAP-CT16 mice was very small. The mean phase delay for CT 16 CNO injection is close to 90 min, while the mean phase delay for 15-minute light pulse at CT 16 is 160 min. However, the mean phase delay for CNO injection at ZT 22 is only 45 min, while ZT 2 is only 33 min. Nevertheless, after adding 2 heterozygous (indicated with thick outlines in Extended Figure 3) to increase the n number to 6, they are still significantly different from GFP control mice injected with CNO.

Additionally, in the TRAP-CT16, CT22 CNO (and, to a somewhat lesser extent, the TRAP-CT16, CT2 CNO) actograms depicted in the author’s response to reviewers that show the best fit lines drawn over the acrophases, the shifts are extremely subtle and do not reflect the best fit lines drawn through the activity onsets in Fig. 1.

The acrophase and activity onset were calculated automatically by ClockLab (Actimetrics, USA). Only a few onsets with small patches of activity near onset time were adjusted according to the overall pattern. The variance of acrophase may be introduced by low activity count in some individuals or injection-induced total activity reduction. However, the overall trend of phase shift is consistent when calculated with acrophase or onset, while the variance between individuals is larger in acrophase. According to acrophase, 4 out of 6 at CT 22 and 5 out of 6 at CT 2 showed clear phase delay after CNO injection. We have added the following description in the result section:

“Analysis of phase shift using acrophase showed that CNO injection at CT 22 or CT 2 can produce phase delay in 4 out of 6 or 5 out of 6 TRAP-CT16 mice respectively (Extended

Figure 3).”

An alternative explanation for the data as presented is that TRAP-CT22 neurons exhibit a light-like phase response to CNO activation (delay when activated at CT 16, advance at CT 22, no change at CT 2). This is consistent with Fig. 1 in the manuscript where the phase marker is activity onset, as well as the figure provided in the response to reviewers where the phase marker is acrophase. However, TRAP-CT16 neurons only exhibit any response when activated by CNO at CT 16. This fits with the figure provided in the response to reviewers (essentially no acrophase phase shift at any time except for CT 16) and in the representative figures in Fig. 1.

We thank the reviewer for providing this alternative explanation to our results. We think it is important to point this out and added this alternative explanation in the discussion section as follows:

“Finally, the phase shifts induced by CNO injection at CT 22 and CT 2 in TRAP-CT16 mice were relatively small compared to those observed at CT 16. There was no significant difference in phase shift between the control group and TRAP-CT16 mice when calculated using the acrophase. Therefore, we cannot entirely rule out the possibility that activating the CT 16 circuit generates a strong phase delay specifically at CT 16 but does not induce a phase shift at other time points. Nonetheless, this alternative explanation still supports our conclusion that the light-activated circuit in the SCN at CT 16 is distinct from that at CT 22.”

A possible explanation for the discrepancy is how the authors calculate phase. As an example, in the representative figure in the response to reviewers for the TRAP-CT16, CT2-CNO, it looks like there may be a mild phase shift between the red line of best fit pre-CNO and blue lines of best fit post-CNO). However, the authors fail to account for the phase advance that occurred on day 5 independent of experimental manipulation. The line of best fit post-CNO therefore looks to be delayed from the line of best fit pre-CNO. If the pre-CNO line of best fit was instead calculated from days 5-8, there would be no phase shift. There appears to be the same issue with the TRAP CT 16, CT2 CNO figure in Fig. 1D, where the pre-CNO onset line of best fit fails to account for the day 5 phase advance, giving a misleading post-CNO onset line of best fit.

Here we follow the suggestion from the reviewer and re-calculate the best fit using stable onset only prior to CNO injection. Indeed, the phase shift is even smaller, but by adding two more heterozygous animals back to statistical analysis, there is still a significant difference between CNO injection at CT 22 and CT 2 in the TRAP-CT16 mice compared to the GFP control mice. In our previous version, we used a longer period

including 5-7 days before CNO injection to draw the best-fit line, and now we have moved that analysis to extended Figure 4. To make it clear for readers, we added specific descriptions in the figure legend as follows:

“c. Statistics of phase shift analysis for vehicle (saline) treatment in the TRAP-CT 16 mice. Here, the phase shift form TRAP-CT16 mice injected with CNO or saline was calculated using onset with best-fit lines generated according to 5-7 days prior to injection.”

I recommend that the authors double-plot their actograms and only draw the best fit lines on one of the plots, as in Grippo et al. 2017 Fig. 2, Jones et al. 2018 Fig. 5, Milosavljevic et al. 2016 Fig. 1, and other papers investigating phase shifts. This will allow the readers to make their best judgment about the extent of the phase shifts in response to CNO activation.

In the current study we generated more than 120 actograms, it may reduce the readability of the manuscript if we provide all actograms. Nevertheless, since TRAP-CT16 injected with CNO at CT 22 and CT 2 are our major phenotypes, we agree with the reviewer and decide to include all 12 actograms from 4 homozygous and 2 heterozygous mice in Extended Figure 3 with double plots according to the reviewer’s suggestion.

I also recommend the authors provide additional representative actograms in the supplemental data. I also still recommend the authors provide an additional reanalysis of the data the data using different phase markers. Even though the authors say the acrophases are unstable in their response to reviewers, if the onsets are not reliable either, it is hard to agree with the figure’s conclusions.

We thank the reviewer’s opinions about the actograms. In the new extended Figure 3, double plots were employed so that the activity onsets were not hidden by notation. Additionally, activity onset and acrophase with their fitting lines were separately marked on each representative actogram for comparison.

Reviewer #2 (Remarks to the Author):

The authors addressed most of my questions, and the manuscript has improved as a result. However, I did not find a response to my last major concern: ‘The paper introduces a dynamic network model with G1, G2, and G3 classifications for SCN neurons. However, the model appears to lack sufficient data to support these classifications. More empirical evidence and statistical validation are needed to strengthen this aspect of the study.’ Considering that the authors have already

addressed my other major concerns, it is acceptable that this issue remains unanswered.

We thank the reviewer for agreeing with most of our previous responses. We want to apologize for missing this comment. The classification and model for our hypothesis are mainly depicted by the conclusion from Figure 6 and Extended Figure 12. Perhaps this is not highlighted sufficiently in the manuscript, so we decided to modify Figure 7 and the legend to emphasize the link between our model and data as follows:

“Proposed bi-stable functional circuit within the SCN comprised with 3 distinct groups of neurons according to Figure 6B (classification), 6C (average light response) and Extended Figure 12 (dynamic response).”

In the original manuscript, we have used two methods to confirm the difference between group 1-3 neurons:

First, between lines 444-446 “When we compare the light response composition, group 1 neurons showed a higher percentage of activation clusters (cluster 1-4) at ZT 16, and group 2 neurons showed a higher percentage of inhibition clusters (cluster 5+6) at ZT 22 (Figure 6D).”

Second, between lines 453-456 “By analyzing the polarity of 18 5-second bins from each ZT, our result revealed distinct patterns for each group in the Haar-ZT space. Interestingly, group 1 was differentiated from groups 2 and 3 at ZT16, whereas group 2 was distinct from groups 1 and 3 at ZT22.”

Here, to further confirm that these three groups of neurons are different from each other, we performed the area under curve (AUC) calculation of their fluorescence response using normalized Z-score (Figure 6C). Statistical analysis confirmed that group 1 neurons showed significantly higher AUC compared to groups 2 and 3 at ZT 16, while group 2 neurons showed significantly lower AUC compared to group 3 at ZT 22. Therefore, light responses from 3 groups of neurons can be separated into at least three classes statistically. We have now added this comparison in the new Extended Figure 11 as the third method to confirm the classification of group 1-3 neurons. The following text was added to the main article:

“To further confirm that these three groups of neurons are different from each other, we performed the area under curve (AUC) calculation of their fluorescence response using normalized Z-scores between 50-75 seconds when the light response was stabilized (Figure 6C). Statistical analysis confirmed that group 1 neurons showed

significantly higher AUC compared to groups 2 and 3 at ZT 16, while group 2 neurons showed significantly lower AUC compared to group 3 at ZT 22. (Extended Figure 11)."

Reviewer #1 (Remarks to the Author):

The authors have satisfactorily addressed my concerns. Thank you for your efforts in greatly improving this manuscript.

Reviewer #2 (Remarks to the Author):

The authors addressed my questions. The manuscript has been largely improved. I have no further questions.

Response: We thank all reviewers for their time, and critical comments which help us improve the manuscript significantly.